# HUST-Grace2024: a new GRACE-only gravity field time series based on more than 20 years satellite geodesy data and a hybrid processing chain

5  Hao Zhou[1,2*], Lijun Zheng[1,2], Yaozong Li[1,2], Xiang Guo[1,2], Zebing Zhou[1,2], Zhicai Luo[1,2]

[1] MOE Key Laboratory of Fundamental Physical Quantities Measurement & Hubei Key Laboratory of Gravitation and Quantum Physics, PGMF and School of Physics, Huazhong University of Science and Technology, Wuhan 430074, P. R. China.

[2] Institute of Geophysics and PGMF, Huazhong University of Science and Technology, Wuhan 430074, P. R. China.

Corresponding author: Hao Zhou (zhouh@hust.edu.cn)

**Abstract.** To improve the accuracy of monthly temporal gravity field models for Gravity Recovery and Climate Experiment (GRACE) and GRACE Follow-On (GRACE-FO) mission, a new series named HUST-Grace2024 is determined based on the updated L1B dataset (GRACE L1B RL03 & GRACE-FO L1B RL04) and the newest atmosphere and ocean de-aliasing product (AOD1B RL07). Compared to the previous HUST temporal gravity field model releases, we made some improvements on both updating background models and processing chain as follows. (1) During the satellite onboard events, the intersatellite pointing angles are calculated to pinpoint the outliers in K-band range rates (KBRRs) and accelerometer observations. To exclude outliers, the advisable threshold is respectively 50 mrad for KBRRs and 20 mrad for accelerations. (2) To relieve the impacts of KBRR noise in different frequencies, a hybrid data weighting method is proposed. Kinematic empirical parameters are used to reduce the low frequency noise, while a stochastic model is designed to relieve the impacts of random noise above 10 mHz. (3) a fully-populated scale factor matrix is used to improve the quality of accelerometer calibration. Analysis in spectral and spatial domain is then implemented, which demonstrates that HUST-Grace2024 has a noticeable reduction of 10% to 30% in noise level and remains consistent amplitudes over 48 basins in signal content compared with the official GRACE and GRACE-FO solutions. These evaluations confirm that our aforementioned efforts lead to a better temporal gravity field series.

## 1 Introduction

The Gravity Recovery and Climate Experiment (GRACE) mission and its follower GRACE-FO (Tapley et al., 2004; Landerer et al., 2020) give us an opportunity to accurately monitor mass transportation for the earth system. The GRACE mission consists of two identical satellites, GRACE-A and GRACE-B, following each other in the same orbital track, linked by a highly accurate inter-satellite K-band microwave ranging system. The outputs of the GRACE mission are a series of monthly temporal gravity field models, represented as a series of harmonic

coefficients for a specific order and degree. The variation in temporal gravity field represented as equivalent water height can be used to monitor the mass transportation for the earth system, including the hydrological cycle over the basins, the variation over the large glaciers, and even extreme weather events like drought or flood events (e.g., Alexander et al., 2016; Amin et al., 2020; Argus et al., 2017; Carlson et al., 2022; Gupta and Dhanya, 2020). In general, the temporal gravity field models give us a new insight into the large-scale mass transportation in the earth system over decades, advancing the progress of geoscience.

The GRACE and GRACE-FO monthly temporal gravity field, denoted as Level 2 (L2) products, is produced from a series of pre-processed Level 1B (L1B) products, which include the GPS observation measured by onboard receivers, the attitude data observed by star cameras, the non-gravitational forces measured by accelerometers, and the range rates derived from the K/Ka band link between two satellites. In order to obtain the temporal gravity field models from these L1B products, a number of background models need to be added during the gravity field determination, which include the ocean tide models, atmosphere and ocean de-aliasing (AOD1B) products, static gravity field and so on. The L2 products are provided by the official solution centers, which include Center for Space Research (CSR, Bettadpur 2018), the German Research Centre for Geosciences (GFZ, Dahle et al., 2018), and the Jet Propulsion Laboratory (JPL, Yuan, 2018). Apart from the official solution centers, there are also a series of research centers computing the monthly gravity field products, which can be found on the website of International Centre for Global Earth Models (Ince et al., 2019).

Due to the limited knowledge of error sources for both observations and background models, the accuracy of current GRACE temporal gravity field models still cannot reach the prelaunch baseline accuracy, which is derived from a pre-launch simulation study (Kim, 2000; Ditmar et al., 2011; Flechtner et al., 2015). In order to improve the quality of temporal gravity field models, many researchers make numerous efforts to figure out the error sources in the observation data and try to model them. For instance, several researchers have discussed the star camera noise and imposed a new combination method based on different star cameras or angular accelerometers in ACC1B, and recomputed the antenna phase center corrections for the inter-satellite observations (e.g., Bandikova and Flury, 2014; Goswami et al., 2018; Horwath et al., 2010). Based on this new processing strategy, the corresponding L1B datasets for star camera attitudes (SCA1B) and K-band ranges (KBR1B) have been updated to release 03 (RL03) in our GRACE solution, while all L1B datasets for the GRACE-FO solution are the newest RL04 edition. Meanwhile, the AOD1B product has been updated from RL05 to RL07 (Dobslaw et al.,

2013; Dobslaw et al., 2017; Shihora et al., 2022), which minimizes the aliasing effect of high frequency mass transportation of the atmosphere and ocean. Apart from updating observation datasets and background models, many agencies also improve their processing chains. Chen et al. (2019) optimized the classical short-arc approach by considering the non-gravitational force errors in ACC1B and attitude errors in SCA1B. Nie et al. (2022) compared the different strategies for force models based on a close-loop stimulation test and gave a new insight into the characteristics of the strategies, which mainly unified the method, including the empirical parameter approach and the filter approach by the Least Squares Collocation (LSC). Abrykosov (2022) developed a self de-aliasing approach for GRACE and GRACE-FO data, which enables us to mitigate the aliasing effect as much as possible.

Meanwhile, the Huazhong University of Science and Technology (HUST) has developed a series of monthly or static gravity field models (Zhou et al., 2017a; Zhou et al., 2019). The previous HUST temporal gravity field has been used in flood event detection (Zhou et al., 2017b) and underground water loss in the North China Plain (Huang et al., 2019; Zhang et al., 2020). The HUST-Grace2020 model also becomes one of the Chinese candidate solutions for the International Combination Service for Time-variable Gravity Fields (COST-G; Meyer et al., 2020). However, all of the previous released temporal gravity field models have not reached the GRACE baseline accuracy (Kim, 2000). Therefore, it is still necessary to determine a series of more accurate temporal gravity field models based on a reasonable processing chain. The motivation for updating the processing chain is to make full use of updated observations and determine more accurate temporal gravity field models. Firstly, we make full use of the sequence of events (SOE) file and intersatellite pointing angles to examine the observation data during the satellite onboard events. The purpose of this step is to examine the relationship between outliers and satellite onboard events, which has not been clearly discussed. Secondly, we construct a hybrid data weighting method for range rate observations, which reduces the low frequency noise following Zhou et al. (2018) and relieve the impacts of high frequency noise via the stochastic model proposed by Ditmar et al. (2006) and Guo et al. (2018). Thirdly, we use the fully-populated scale factor matrix to improve the quality of accelerometer calibration (Klinger and Mayer-Gürr, 2016). Based on the updated processing chain, a new series of temporal gravity field models named HUST-Grace2024 is determined. The models span from April 2002 to June 2017 for GRACE and from June 2018 to December 2022 for GRACE-FO, which almost cover the whole observational period for GRACE and GRACE-FO mission. Compared to the recent RL06 products from the GRACE official solution centers, a

clearly temporal noise reduction of about 20% over a selected open ocean can be found in HUST-Grace2024, while annual amplitude is still consistent with the official solutions, which supports the necessity of updating the processing chain for our temporal gravity field determination.

This paper is structured as follow: Section 2 presents the updating processing chain for HUST-Grace2024. Section 3 presents the quality of HUST-Grace2024 model in both spectral and spatial domain. Section 4 is for conclusion.

## 2 Methods and GRACE data processing

### 2.1 Updating of GRACE data and background models

The classical dynamic method has been successfully used for recovering temporal or static gravity field model series by different agencies (Reigber, 1989; Dahle et al., 2018; Mayer-Gürr et al., 2018). The dynamic approach takes the gravity field inversion problem as a variation equation, including initial state and force model parameters. Based on the classical dynamic method, we have developed a software platform to determine the gravity field product series, including HUST-Grace2016s, HUST-Grace2016, HUST-Grace2019 and HUST-Grace2020 (Zhou et al., 2017a; Zhou et al., 2019). Using this software platform as well as the reprocessed latest GRACE/GRACE-FO L1B data and the updating data processing chain (Table 1), a new GRACE and GRACE-FO temporal gravity field series named HUST-Grace2024 is then developed. Compared to previous HUST products, we made several improvements in terms of the updated observation datasets and the newly released background force models. Specifically, the improvements include (1) using the latest L1B data based on some improved processing strategies implemented by the L1B data processing center; (2) using the new ITSG kinematic products to obtain more accurate satellite positions; (3) updating background force models, including GOCO06s and a new atmosphere and ocean de-aliasing product (AOD1B RL07) to remove additional sub-monthly or higher frequency temporal signals.

Apart from the newest observation datasets and background force models, we also updated the processing chain. In the following parts of this section, some aspects of the HUST-Grace2024 processing chain will be described in detail, including (1) a new HUST-Grace2024 data pre-processing strategy, (2) a hybrid data weighting method, and (3) a fully populated scale matrix for GRACE accelerometer calibration. It's necessary to stress that the sample rate for kinematic is different with the other L1B data. So, we simply we simply truncate the original

observation equation for integration orbit according to the GPS time tag in the kinematic observation. Actually, during the kinematic preprocessing, we use the reduced dynamic orbit as the criteria for error identification, and when the difference between the reduced dynamic orbit and the kinematic orbit exceeds 20 cm, we will give a quality flag to the kinematic orbit at a specific GPS time and will not use the kinematic observation for the temporal gravity field determination later on. As for the gap in the kinematic observation, we fill the gap by zero value and don't use the observation to construct the observation equation.

**Table 1 Summary of input data, reference frames, force models and estimated parameters for temporal gravity field model determination.**

| Input Data | Description |
| --- | --- |
| Kinematic orbits | ITSG products (Strasser et al., 2018), 10 seconds sampling |
| K-Band range-rates | KBR1B, Level 1B, 5 seconds sampling[*] |
| Attitude observations | SCA1B, Level 1B, 5 seconds sampling[*] |
| Accelerometer observations | ACC1B, Level 1B, 5 seconds sampling[*] |
| **Reference Frames** | |
| Conventional inertial reference system | IERS Convention 2010 (Petit & Luzum, 2010) |
| Processing and nutation | IAU 2006/2000A (Petit & Luzum, 2010) |
| Earth orientation parameters | IERS EOP 14 C04 |
| **Force Models** | |
| Earth's static gravity field | GOCO06s (Kvas et al., 2019) truncated to degree/order 180 for static part while trend part and oscillation part truncated to degree/order 120 |
| Ocean tide | EOT11a (Savcenko et al., 2012), truncated to degree/order 120, including 18 main waves and 234 secondary waves |
| N-body Perturbation | JPL DE421 (Folkner et al., 2009), Sun and Moon, direct and indirect terms, indirect J2 Effect |
| Solid earth Tide | IERS Conventions 2010, include frequency independent term, frequency dependent term and permanent tide |
| Solid earth pole tide | IERS Conventions 2010, $C_{21}$ and $S_{21}$ |

| | |
|---|---|
| Ocean pole tide | Desai (Desai, 2002), truncated to degree/order 100 |
| Atmosphere and oceanic variability | AOD1B RL07 (Shihora et al., 2022), truncated to degree/order 180, linear interpolation, air tide S2 was considered |
| General relativistic effect | IERS Conventions 2010, Sun and Earth |
| **Estimated parameters** | |
| Initial state vector | Once per arc, include 6 parameters per satellite |
| Accelerometer parameters | Biases, once per arc (6 hours), include biases in X, Y, Z components, include quadratic polynomial |
| | Scales, once per month, full scale matrix |
| Range-rate empirical parameters | Once per 5400s, include bias, slope and 1 cycle-per-revolution (1-CPR) |
| Geopotential coefficients | Complete to degree and order 96 |

\* KBR RL03 and SCA RL03 data spans from 2002.04 to 2017.06, ACC RL02 data spans from 2002.04 to 2016.10 and its RL03 data spans from 2016.11 to 2017.06 due to GRACE-B ACC data filled by transplant approach based on the data from GRACE-A ACC data for GRACE data. GRACE-FO L1B RL04 data including ACH1B, SCA1B, and KBR1B spans from 2018.06 to 2022.12.

It should keep in mind that choice of kinematic orbit or the reduced dynamic orbit derived from may also have impact on temporal gravity field determination. In order to make a quantitative for this impact, we design a control variable experiment as follow: (1) the GNV1B product is resampled by 10 seconds, consistent with the sample rate of kinematic; (2) the noise for both orbit and rang-rate observation is regarded as white noise, and the accuracy for orbit observation used for weight factor determination is 2 cm while the range-rate observation is 0.2 μm/s. (3) the computation finished at degree and order 120. As shown in Figure 1, the temporal gravity field is determined based on the GNV1B and kinematic products. The variation of geoid height difference computed by GNV1B is similar to that computed by kinematic below degree 40. However, the geoid height difference computed by GNV1B is generally larger than that computed by kinematic at degree 40 to 60, degree 110 to 120. In order to make a quantitative assessment for the impact from GNV1B or kinematic, we calculate the cumulative geoid height difference for the temporal gravity field products. The cumulative geoid height difference is 8.89 cm, 9.24 cm for kinematic and GNV1B respectively. The result indicates that the adoption of kinematic has a potential

positive effect on temporal gravity field. It's advisable to replace the reduced dynamic orbit by the kinematic orbit during the temporal gravity field determination.

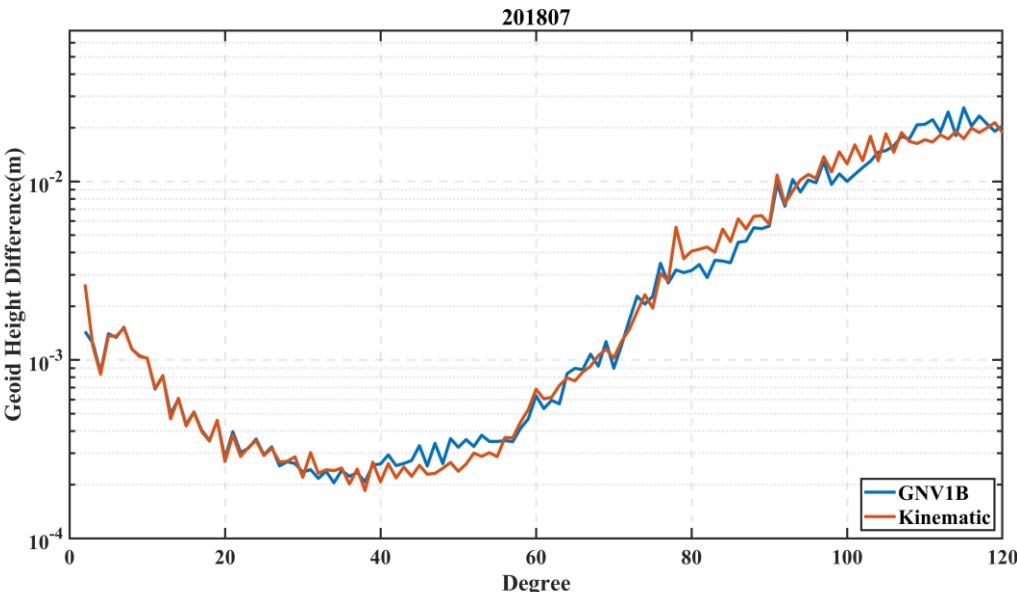

**Figure 1 Geoid Height Difference computed by different orbit products, the blue curve is computed by the reduced dynamic orbit derived from GNV1B and the red curve is computed by the kinematic orbit.**

## 2.2 Renewing of HUST-GRACE2024 data pre-processing strategy

During the temporal gravity field determination, observation data pre-processing plays an extremely important role in excluding outliers from the final products used in the recovery process. Generally, the data pre-processing procedure includes: (1) filling some data gaps by using interpolation value; and (2) excluding some large outliers due to satellite onboard events. For the first aspect, we fill the data gaps in ACC1B and SCA1B by interpolation values, following the method mentioned in the GRACE L1B user handbook (Case et al., 2010). As for the second aspect, we not only exclude the outliers by the data flags but also reject the degraded data, which is associated with satellite onboard events according to the sequence of events (SOE) files. For outliers in the observation data during temporal gravity field determination, most processing centers exclude them based on the residual value (reference value minus observation data), which is based on the observation instrument data itself. However, in HUST-Grace2024, we make use of multi-observation data to determine outliers based on cross validation.

There are several kinds of satellite onboard events: center of mass (CoM) calibration maneuvers, satellite battery management maneuvers, satellite swap position maneuvers, orbital maneuvers, and so on. Some events, such as

the CoM calibration maneuvers, can be found in the SOE file, which will be updated if necessary. For these events founded in the SOE file, we add an addition flag into the final pre-process products, which will not be used in the following gravity field recovery chains. For instance, the accelerometer data is tagged during the CoM calibration maneuvers and the corresponding orbital observation is discarded from the scale factor determination, finally maintaining a stable monthly scale factor. Moreover, other events such as satellite battery management or satellite swap maneuvers are not recorded in the SOE file, which may also affect the observation quality and contaminate the final gravity solution. Therefore, it is necessary to figure out these events according to the special patterns in the observation data.

To figure out the non-recorded events in the SOE file, a line-of-sight (LOS) Euler angle variation between satellites-based SCA1B and GNV1B data had been developed. Based on the method, satellite swap maneuvers and their related events can be detected. The core of the method is to find the line-of-sight Euler angle variation. The calculation works as follow: Firstly, let us denote the rotation matrix from inertial reference frame (IRF) to K-Band frame (KF) as $R_{IRF}^{KF_j}$, the matrix from IRF to LOS as $R_{IRF}^{LOS_J}$, and the matrix from IRF to science reference frame SRF as $R_{IRF}^{SRF_j}$, respectively. The matrixes can be estimated as follow.

$$R_{IRF}^{SRF_j} = \begin{bmatrix} x_{SRF_j}^T \\ y_{SRF_j}^T \\ z_{SRF_j}^T \end{bmatrix} = \begin{bmatrix} q_0^2 + q_1^2 - q_2^2 - q_3^2 & 2*(q_1*q_2 + q_0*q_3) & 2*(q_1*q_3 - q_0*q_2) \\ 2*(q_1*q_2 - q_0*q_3) & q_0^2 - q_1^2 + q_2^2 - q_3^2 & 2*(q_2*q_3 + q_0*q_1) \\ 2*(q_1*q_3 + q_0*q_2) & 2*(q_2*q_3 - q_0*q_1) & q_0^2 - q_1^2 - q_2^2 + q_3^2 \end{bmatrix} \quad (1)$$

$$R_{IRF}^{KF_j} = \begin{bmatrix} x_{KF_j}^T \\ y_{KF_j}^T \\ z_{KF_j}^T \end{bmatrix} = \begin{bmatrix} \left(R_{IRF}^{SRF_j}\right)^T \left(\dfrac{phc}{|phc|}\right)^T \hat{e}_{IRF}^{SRF} \\ \left(z_{KF_j} \times x_{KF_j}\right)^T \\ \left(\left(\left(\dfrac{phc}{|phc|}\right)\left(R_{IRF}^{SRF_j}\right)\hat{e}_{IRF}^{SRF}\right) \times y_{SRF_j}\right)^T \end{bmatrix} \quad (2)$$

$$R_{IRF}^{LOS_j} = \begin{bmatrix} x_{LOS_j}^T \\ y_{LOS_j}^T \\ z_{LOS_j}^T \end{bmatrix} = \begin{bmatrix} \left( \dfrac{r_i - r_j}{|r_i - r_j|} \right)^T \\ \left( \left( \dfrac{r_i - r_j}{|r_i - r_j|} \right) \times \dfrac{r_i}{|r_i|} \right)^T \\ \left( x_{LOS_j} \times y_{LOS_j} \right)^T \end{bmatrix} \tag{3}$$

where $r_j$ or $r_i$ denotes the satellite baseline vector ($i \neq j$). $phc$ denotes the phase center, which can be obtained

from VKB1B. $q_0, q_1, q_2, q_3$ are the Quaternion numbers, which can be derived from SCA1B. $\hat{e}_{IRF}^{SRF}$ is the unit

vector from IRF frame to SRF frame. Index $j$ is satellite ID.

Secondly, substitute equation (2) and equation (3) into equation (4) as follow.

$$R_{KF}^{LOS} = R_{IRF}^{LOS} \left( R_{IRF}^{KF} \right)^T \tag{4}$$

where $j$ is omitted for a better demonstration. Note that $R_{KF}^{LOS}$ is a rotation matrix for transforming from KF to

LOS. Thus, the rotation matrix can be divided into three independent directions as follow.

$$R_{KF}^{LOS} = R_1(\phi) R_2(\theta) R_3(\psi) \tag{5}$$

where ϕ, θ, ψ is Euler pointing angles named roll, pitch and yaw, respectively.

Based on the aforementioned method, the Euler angle variations for 2011.09.06 and 2013.01.04 are shown in

Figure 2. It can be clearly seen that there is a huge data gap in KBR range-rate data lasting from 14:00:00 to

14:45:00. At the same time, there is a notable oscillation in yaw-angle variation data. Thus, there are some

connections between range-rate data loss and yaw-angle change. When yaw-angle exceeds a threshold, for

instance, 50 mrad, there is a data gap in range-rate data.

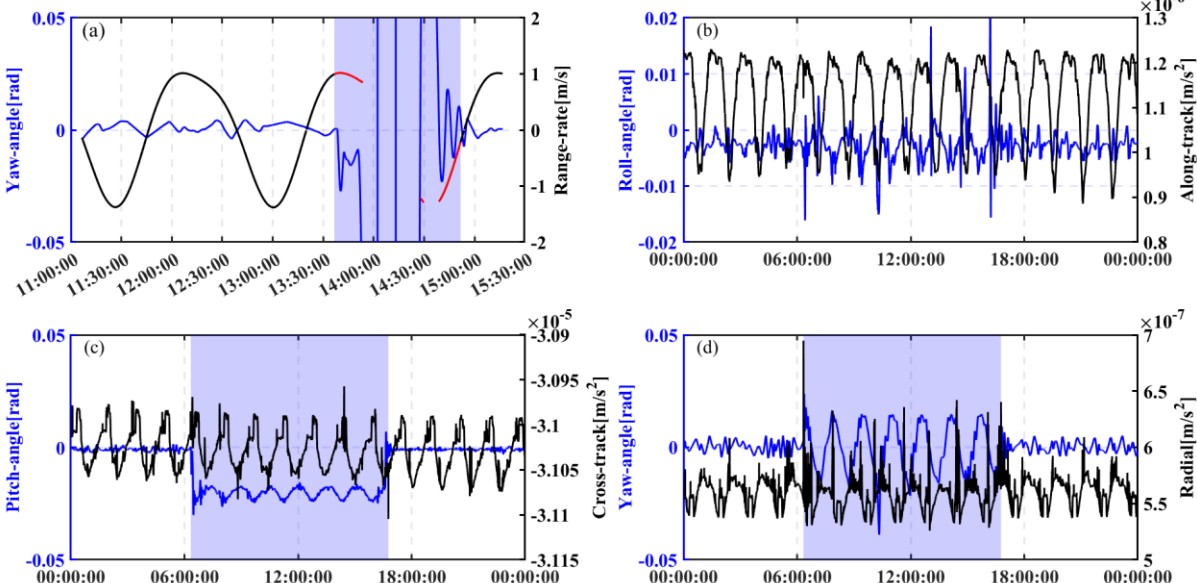

**Figure 2 (a) Time series of GRACE-A yaw pointing angles variations (in blue lines) during yaw-turn maneuvers in 2011.09.06. The KBR range rates are plotted in black lines, and the screened data is shown in red. The KBR observations for the time period highlighted in light-blue will not be used in the following temporal gravity field recovery process. (b to d) Time series of Euler pointing angles variations (in blue lines) during CoM calibration maneuvers in 2013-01-04. The ACC observations in along-track, cross-track and radial for GRACE-A are also shown as black lines.**

Comparing the Euler angles as shown in Figure 2 (a), we set 50 mrad as the threshold in yaw-angle variation and 100 seconds as the margin interval. That is, when the Euler angles exceed the threshold, the corresponding range rates in the margin interval will be excluded from the following gravity field recovery process. To illustrate the necessity of excluding these observations, as shown in Figure 3, the geoid height variance per degree is also computed via different pre-processing strategies (including original-processing and updated-processing strategies). In 2005.01, there were also some yaw-turn maneuvers like Figure 2 (a), which degraded the quality of KBR range rate data. In the updating-processing strategy, we exclude the KBR range rate data in the corresponding time based on the aforementioned outlier detection method. Compared to the original-processing strategy, the updated-preprocessing strategy has a relatively smaller geoid height difference per degree than the original one from degree 35 to 90, with a reduction of about 13% at most. It clearly illustrates the benefit of updating the pre-processing strategy.

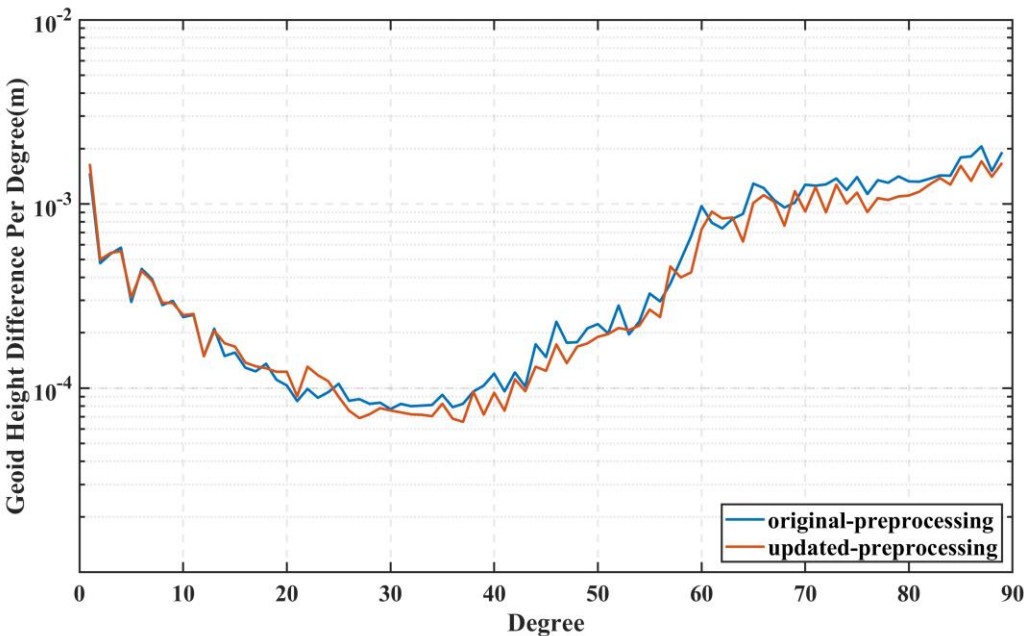

**Figure 3 The geoid height variance of gravity fields in 2005.01 computed by original-processing strategy (in red lines) and updated-processing strategy (in green lines).**

During the CoM calibration maneuvers, as shown in Figure 2 (b to d), the pointing angle, especially the yaw-angle change in a sine-like pattern with an amplitude of about 20 mrad. In this study, according to the SOE file, the CoM calibration maneuvers are executed at the same time. It is reasonable since the CoM offset will be set to constant during calibration maneuvers, lasting about 180 seconds. During this time, the linear acceleration in three directions will suffer from a regular square wave variation without additional thruster firing. Actually, only a start-time tag and a finish-time tag is recorded during CoM calibration. Based on this method, in HUST-Grace2024 gravity field series, SOE file is used for picking out the calibration campaign time tag and the Euler angles variation is also computed. The degraded data in specific arc is excluded during monthly scale factor estimation and gravity field determination.

## 2.3 Hybrid data weighting method

In this section, the second improvement of the processing chain in HUST-Grace2024 is discussed, which mainly focuses on developing a hybrid data weighting method in gravity field determination. Except for the systematic errors, which can be physically corrected beforehand, there is still some random noise in the observations.

Therefore, it is necessary to construct a proper hybrid weighting method, which can comprehensively process the systematic errors and the random noise.

The estimation of geo-potential coefficients can be described as follow.

$$x = (A_{orbit}^T P_{orbit} A_{orbit} + A_{kbr}^T P_{kbr} A_{kbr})^{-1}(A_{orbit}^T P_{orbit} l_{orbit} + A_{kbr}^T P_{kbr} l_{kbr}) \tag{6}$$

where $x$, $A_{orbit}$, $P_{orbit}$, $l_{orbit}$ denotes the unknown geo-potential coefficients, design matrix for satellites orbit, stochastic model for orbit, orbit residual vector computed by integration orbit minus observation orbit derived from L1B data, respectively. $A_{kbr}$, $P_{kbr}$, $l_{kbr}$ denotes design matrix, stochastic model, residual vector for KBR observation data, respectively.

For the $P_{kbr}$, the stochastic model is constructed by post-fit range rate residuals. As shown in Figure 4 (a), the KBR range rate prefit residuals suffer from low-frequency noise (<1mHz) associated with satellite orbit, where its relationship can be described by Hill equation (Colombo, 1984) and high frequency random noise (>10mHz).

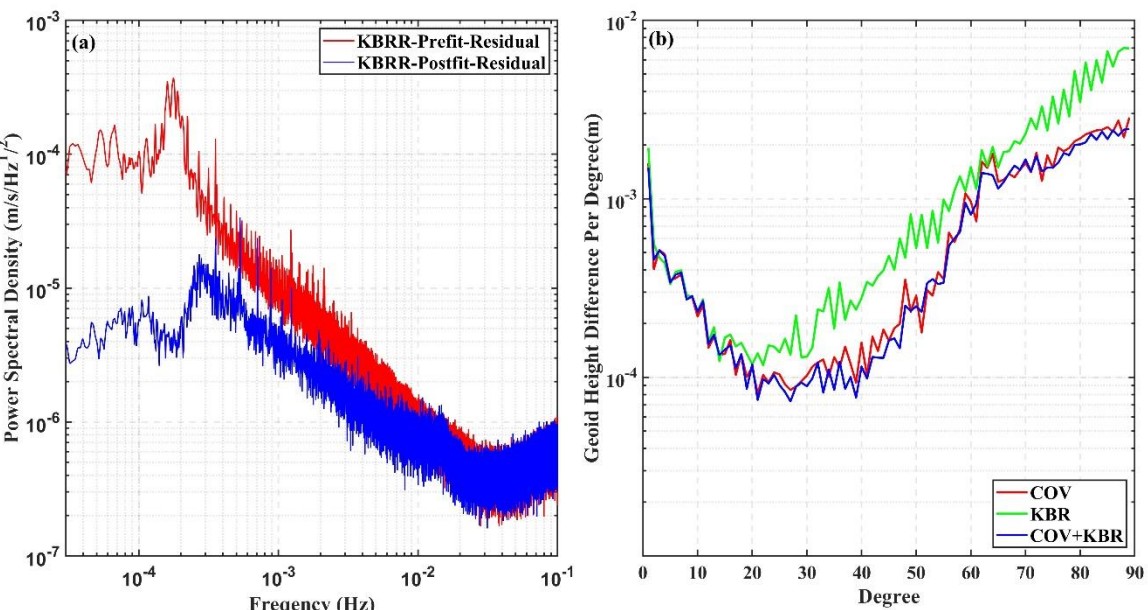

**Figure 4 (a) Power Spectral Density (PSD) of one month GRACE KBRR range rates in 2005-01. The red line is computed by KBRR prefit residuals while the blue line is computed by KBRR postfit residuals. (b) The geoid height difference of gravity fields in 2005.01 computed by different low frequency noise processing strategies including KBRR data via COV (red line), KBR (in green line), and KBR+COV (in blue line), where COV denotes covariance stochastic model matrix, KBR denotes range-rate empirical parameters, COV+KBR denotes hybrid preprocessing strategy including COV and KBR method.**

For the low-frequency noise in range rate data, some range rate empirical parameters are commonly used in low-frequency noise reduction (Zhou et al., 2018; Nie et al., 2021). However, the high-frequency random noise cannot be reduced as much as the low-frequency one. As shown in Figure 4 (a), it is noticeable that some gravity signals can be derived from KBRR pre-fit residuals, while the temporal signals can be hardly retrieved from KBRR pre-fit residuals over 10mHz. In order to retrieve as many gravity signals as possible, the correlations among range rate observations must be taken into consideration via a stochastic model.

Based on the method proposed by Ditmar (2006), the covariance matrix $P_{kbr}$ is constructed by the auto-covariance vector, which is originated from the post-fit residuals of KBR observations. The steps of constructing a covariance matrix can be summarized as follows.

(a) computing the auto-covariance vector for post-fit range rate residual series as

$$a_k = \frac{1}{N_k} \sum_i s_i s_{i+k} (0 \leq i \leq n) \tag{7}$$

where $a_k$ denotes $k$-$th$ auto-covariance vector element, $N_k$ denotes the pairs of post-fit residual elements used for the auto-covariance elements estimation, and $i$ denotes the lag between different epochs in residual series. It should keep in mind that $k$ must be in the interval $(0, n)$, where $n$ is the total number of residual series.

(b) After calculating the auto-covariance vector, the Toeplitz covariance matrix is then constructed as follow.

$$\begin{pmatrix} a_0 & \cdots & a_{n-1} & a_n \\ \vdots & a_0 & \cdots & a_{n-1} \\ a_{n-1} & \cdots & \ddots & \vdots \\ a_n & a_{n-1} & \cdots & a_0 \end{pmatrix} \tag{8}$$

Following the aforementioned methods, we design an experiment to investigate the effect of different strategies for KBR observations in real GRACE gravity field determination, as shown in Figure 4 (b). During the experiment, the strategies via the range-rate empirical parameters (KBR in Figure 4 (b)), the covariance matrix (COV in Figure 4 (b)), and the hybrid data weighting approach COV+KBR are applied. It is obvious that all strategies have almost the same long-wavelength gravity signal capture ability, while the estimation via the hybrid approach COV+KBR presents the smallest geoid height variance. There are still some results that should be noticed: (1) The COV approach is noticeably different from the KBR one, especially for the estimations after degree 20. (2) The result

of the hybrid approach is slightly better than the COV one between degree 25 and 50. The results in Figure 4 (b) give us an insight into the characteristics of the different KBR data processing strategies.

For the first part of the result, the relationship between the COV method and the KBR method has been described by Nie (2022), which can be united by the least squares collocation. However, the results in Figure 4 (b) demonstrate that there is still some difference between the COV method and the KBR method for real data processing. To qualify the difference, the gravity field determination equation (6) can be rewritten as follows:

$$x = (A_{orbit}^T P_{orbit} A_{orbit} + A_{kbr}^T F^T Q_{kbr} F A_{kbr})^{-1} (A_{orbit}^T P_{orbit} l_{orbit} + A_{kbr}^T F^T Q_{kbr} F l_{kbr}) \tag{9}$$

where $F$ denotes a linear operator, $Q_{kbr}$ implies the stochastic model matrix. If the result of gravity field determination remains the same, the COV and KBR method must meet the equation (10).

$$P_{kbr} = F^T Q_{kbr} F \tag{10}$$

However, the equation (10) can hardly be met in real gravity field determination due to the operator characteristic of $F$. Different from the KBR method, the COV method is constructed based on the correlation of range rate observations, including all signals in the target frequency band. Thus, the COV method results in better gravity field estimations than the KBR method.

In conclusion, the COV method mainly attenuates the effect of random noise in range rate observations during gravity field determination, while the KBR method mainly attenuates the systematic effect from orbit. Therefore, our hybrid data weighting method combining COV with KBR can achieve a better result than the individual method.

**2.4 Strategy for accelerometer calibration**

Generally, the raw ACC1B measurement data is degraded by unknown scale factors, time-varying biases, and other unknown noise, so the data need to be calibrated by the equation (11) before the gravity field determination.

$$a_{cal} = R * (S * a_{obs} + b_{bias}) \tag{11}$$

$$S = \begin{bmatrix} s_x & \alpha + \zeta & \beta - \varepsilon \\ \alpha - \zeta & s_y & \gamma + \delta \\ \beta + \varepsilon & \gamma - \delta & s_z \end{bmatrix} \tag{12}$$

where $a_{obs}$ denotes the non-gravitational forces observed by the onboard accelerometer, $S$ denotes the scale factor matrix, $b_{bias}$ implies the bias vector for calibration, $R$ implies the rotation matrix transforming the accelerometer observations from SRF to IRF, and $a_{cal}$ denotes the calibrated accelerometer measurements. It should be noted that there are three parts in the scale factor matrix $S$: (1) main diagonal elements consisting of $s_x$, $s_y$ and $s_z$, (2) shear elements consisting of $\alpha$, $\beta$ and $\gamma$, (3) rotational elements including $\zeta$, $\varepsilon$ and $\delta$.

As shown in Table 1, the following accelerometer calibration parameters for HUST-Grace2024 are estimated. (1) a monthly fully-populated scale factor matrix including main diagonal and off-diagonal elements, (2) a 6-hour bias vector including constant, first time-derivates, and second time-derivates bias elements for along-track, cross-track, and radial directions. Based on the calibration Equation (11), we estimated the scale factor matrix, as shown in Figure 5. The scale factor variation is similar for both GRACE-A and GRACE-B, thus only the parameters for GRACE-A are plotted in this figure.

The main diagonal element at cross-track direction is not estimated to be as stable as those at the other two directions, which is derived from the limited sense of the accelerometer at cross-track direction. Moreover, for the radial axis, the main diagonal scale factors fluctuate more violently in 2002 to 2006 and 2012 to 2017 than those in 2007 to 2011. The violent fluctuation may be associated with the time period of solar activity, which can be supported by Koch et al. (2019). When the solar activity is serene, for instance, the scale factor at the radial direction can be estimated stably during 2006 to 2010, while it cannot be estimated as stable during 2012 to 2017. A clearly reduced along-track scale factor can be seen after 2011 due to the switched-off thermal control for GRACE-A. Meanwhile, the scale factor in the radial direction is more and more scattered as the GRACE mission moves on. The results indicate that the scale factor has a clearly reduced behaviour itself, which is consistent with the results as shown in Klinger and Mayer-Gürr (2016). When comparing with the main diagonal estimations derived from GRACE, those derived from GRACE-FO are notably scattered, especially for the along-track estimations. The scattered estimations are mainly due to the relatively degraded performance of GRACE-FO onboard accelerometers.

As for the rotation elements and shear elements, they are not presented as constants but vary with very similar patterns, which indicates that the misalignment between SRF and accelerometer frame (AF) has a close connection with the non-orthogonality of three AF axes. In addition, there is general comparable performance between GRACE and GRACE-FO estimations, demonstrating the comparable alignment accuracy for SRF and AF as well as the orthogonality of three AF axes.

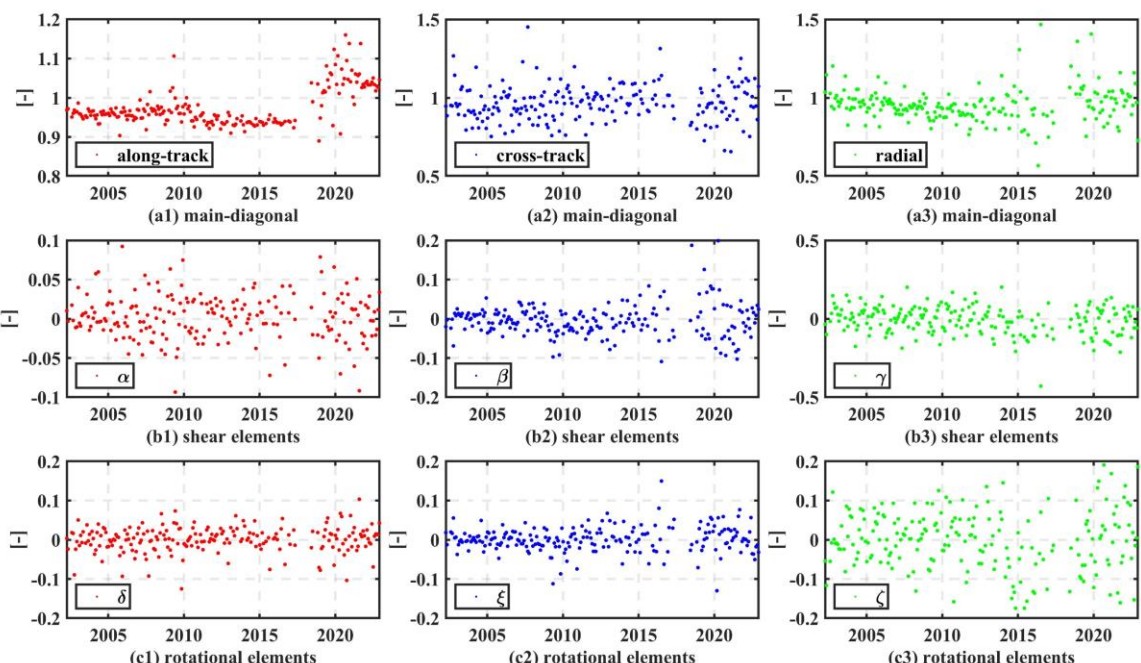

**Figure 5 Full calibration scale factor elements for GRACE-A in monthly resolution. (a1-a3) main diagonal elements in along-track (red), cross-track (blue) and radial (green) direction, respectively. (b1-b3) shear elements variation of α (red), β (blue), γ (green). (c1-c3) rotational elements variation of δ (red), ξ (blue), ζ (green).**

**Table 2 Mean value and standard deviation of accelerometer scale factors for GRACE-A, GRACE-B during 2002.04 to 2017.06. The corresponding prior values from TN02 are also listed (Bettadpur, 2009).**

|  | GRACE-A | GRACE-B | TN02-A | TN02-B |
|---|---|---|---|---|
| along-track | 0.954 ± 0.022 | 0.942 ± 0.017 | 0.960 ± 0.002 | 0.947 ± 0.002 |
| cross-track | 0.968 ± 0.110 | 0.964 ± 0.108 | 0.980 ± 0.020 | 0.984 ± 0.020 |
| radial | 0.942 ± 0.127 | 0.924 ± 0.102 | 0.949 ± 0.020 | 0.930 ± 0.020 |
| α | 0.001 ± 0.029 | 0.003 ± 0.026 | - | - |
| β | -0.006 ± 0.031 | 0.006 ± 0.032 | - | - |
| γ | -0.012 ± 0.106 | 0.003 ± 0.094 | - | - |

| | | | | |
|---|---|---|---|---|
| ε | -0.000 ± 0.030 | 0.001 ± 0.026 | - | - |
| ξ | 0.000 ± 0.031 | 0.005 ± 0.031 | - | - |
| ζ | 0.011 ± 0.079 | 0.012 ± 0.082 | - | - |

For a general comparison of the scale factors during the whole GRACE lifetime, we compute the corresponding mean values and standard deviations for GRACE-A and GRACE-B. Meanwhile, the prior values of the main diagonal elements from TN02 are also listed in Table 2. The mean values of the scale factor at along-track, cross-track, and radial directions estimated for HUST-Grace2024 are close to the corresponding prior values derived from TN02, which demonstrates the stability of our accelerometer parameterization strategy. Note that, although these mean values for HUST-Grace2024 have a close connection with TN02 (not exactly the same), the scale factors still need to be estimated again along with the geopotential coefficients to avoid the potential temporal signal attenuation (Zhou et al., 2019). Except for the main diagonal elements, we also compute the mean values of rotational elements and shear elements. According to Table 2, the mean values are quite small (not zero), while the relevant standard deviations are still noteworthy when compared to those for the main diagonal. It indicates the necessity of estimating rotational elements and shear elements, which support the thought of the misalignment between SRF and AF and the non-orthogonality of three AF axes.

Overall, our hybrid processing chain contribution to HUST-Grace2024 can be summarized as follows: (1) updating the observations, background force models and key parameter estimation strategies for the whole observation period of GRACE and GRACE-FO mission. Meanwhile, based on the fully populated scale factor matrix, we obtain the relatively stable accelerometer scale factors. Compared with GRACE estimations, the accelerometer scale factors of GRACE-FO mission fluctuate more widely. (2) making full use of the inter-satellite pointing angles to detect outliers in accelerometer and range rate data, which is different from the previous studies and other processing centers outlier detection methods. In addition, we make a quantitative analysis related to the impacts of our outlier detection method on temporal gravity field determination, which can be also an important instruction to quantitatively analyse the effect of satellite onboard events in the data processing chain. (3) making a comprehensively quantitative analysis of different weighting approaches for frequency-dependent noise in range rates, and further analysing their impacts on temporal gravity field determination.

## 3 GRACE results

### 3.1 HUST-Grace2024 Model

Based on the updated processing chain, a new temporal gravity field series named HUST-Grace2024 during the whole GRACE and GRACE-FO science mission lifetime from 2002 to 2022 has been developed. To evaluate the performance of HUST-Grace2024, we make a comprehensive comparison with the latest official gravity field solutions, including CSR, GFZ, and JPL, in both the spectral and spatial domain. The time span for gravity field solutions assessment is selected from 2005 to 2016 for GRACE solutions and 2018 to 2022 for GRACE-FO solutions.

### 3.1.1 Analysis in Spectral Domain

A commonly used parameter to quantify the quality of GRACE solutions in the spectral domain is the geoid height variance compared to a static gravity field. In this section, the static field EIGEN-6C4 is used as the base gravity field. The geoid height difference is dominated by the temporal gravity signal for low degree geopotential coefficients, while it is dominated by the mismodeling gravity signal for high degrees. Therefore, the smaller geoid height variance per degree at high degrees over 40 can be used as a criterion for evaluating the GRACE solutions in the spectral domain.

As shown in Figure 6, the geoid height difference per degree of HUST-Grace2024 is compared with three official solutions, including CSR RL06, GFZ RL06, and JPL RL06 (RL06.1 for GRACE-FO solutions). Generally, the HUST-Grace2024 has almost the same or slightly better ability to retain long-wavelength signals in the temporal gravity field, while there is a noticeable noise reduction of geopotential coefficients at high degrees. Especially, as shown in Figure 6 (a to f), the geoid height difference per degree of HUST-Grace2024 is smaller than those of the official solutions from degree 40 to degree 80, indicating a better performance of the HUST-Grace2024 model in capturing high frequency mass transportation.

For a better comparison of the spectral characteristics for HUST-Grace2024, as listed in Table 4, we also computed the cumulative geoid height difference for HUST-Grace2024 and three official solutions. During 2005 to 2010, the average cumulative difference (*cm*) was respectively 0.75, 0.98, 1.00, and 0.66 for CSR RL06, GFZ RL06, JPL RL06, and HUST-Grace2024, indicating a average cumulative geoid height difference reduction of -12.8%,

-33.2%, and -34.7% for the HUST-Grace2024 model when compared to the official solutions. Especially, in August 2006, the reduction for HUST-Grace2024 reached 45.0% and 42.6% when compared to GFZ RL06 and JPL RL06. As for the GRACE-FO mission period, the noise reductions for HUST-Grace2024 become more notable when compared to the official solutions. The results indicate our hybrid processing strategy is likely more efficient for the GRACE-FO observations. The reduction of the geoid height difference may be contributed by the degrees up to 40. In order to assess the temporal signal in HUST-Grace2024, we need to make an analysis in the spatial domain as described in the following section.

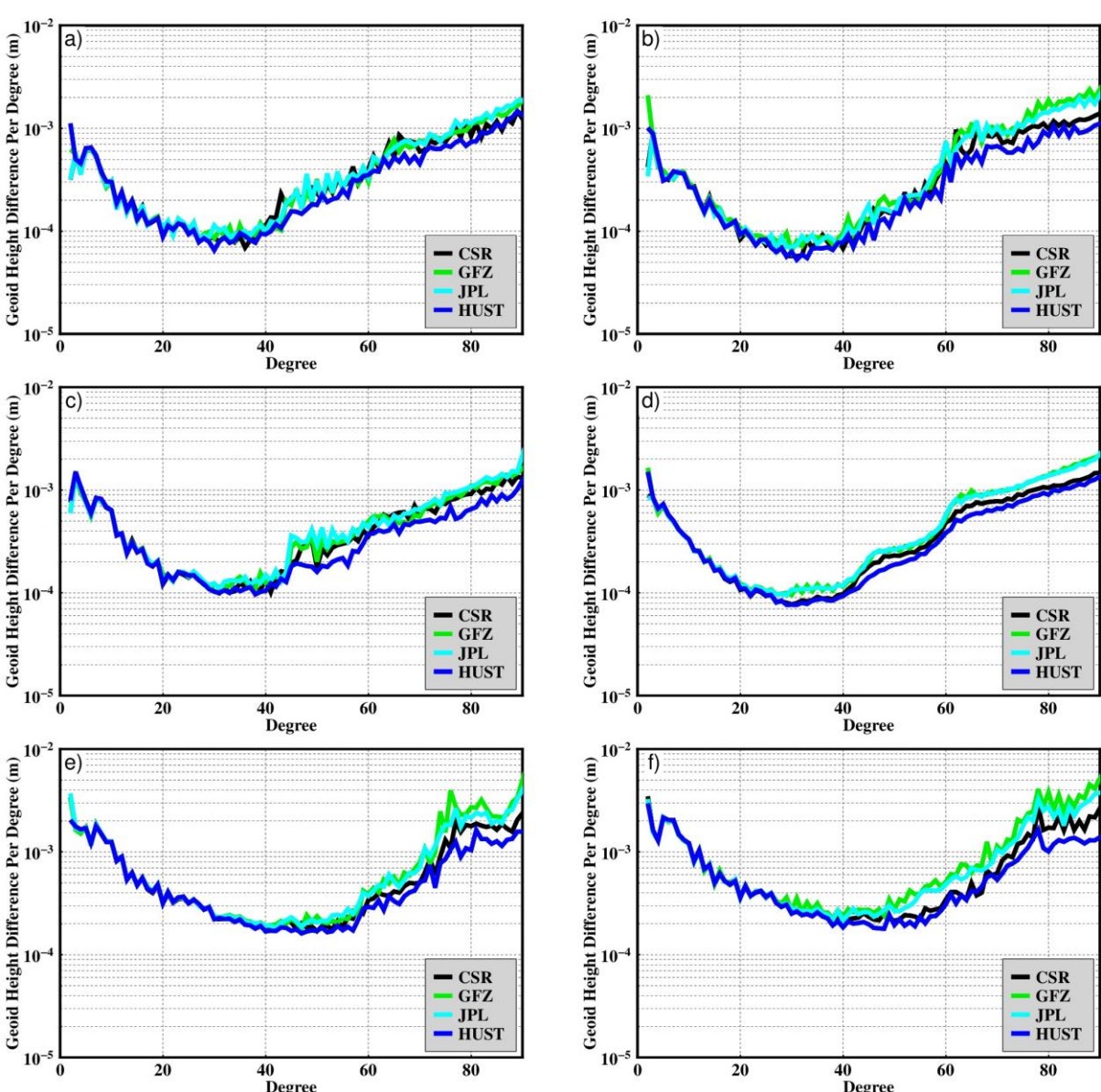

**Figure 6 The geoid height difference per degree of temporal gravity model at different months including (a) 2009.01, (b) 2006.08, (c) 2010.10, (d) 2005 to 2010, (e) 2019.10, (f) 2022.05.**

$C_{20}$ has the largest magnitude among the geopotential coefficients, which has a close connection with the Earth's geodynamic process and large-scale mass transportation (e.g., Cheng and Ries, 2023; Landerer et al., 2020; Velicogna et al., 2020). It is very important to obtain the accurate $C_{20}$ estimation during the temporal gravity field determination. As shown in Figure 2, the non-gravitational signal observed by the accelerometer has a periodic variation, such as 1-CPR and 2-CPR later on. The non-gravitational feature can be captured by the resonant orders in geopotential coefficient. The satellite laser ranging (SLR) tracking consists of several geodetic satellites such as Starlette, Stella, and Ajisai, which are operated at different altitudes and experience different "lump effects" of geopotential coefficients. Thanks to the characteristics of SLR, zonal coefficients derived from SLR are more reliable than those derived from GRACE (Cheng and Ries, 2017; 2023). Therefore, we compare these geopotential coefficients from GRACE solutions with those derived from SLR, qualifying the accelerometer calibration method indirectly. Besides this, the $C_{20}$ and $C_{30}$ are degraded and corrupted by the thermal noise in the accelerometer and a 161-day periodic signal, and the values need to be replaced by SLR, especially after the thermal control was switched off in 2011 during the GRACE science mission.

As shown in Figure 7, generally, the variation of $C_{20}$ derived from HUST matches well with that derived from SLR. The correlation coefficient for $C_{20}$ with SLR is also computed, as listed in Table 3. It is clearly implied that the correlation coefficient between HUST and SLR is 0.86, and the other SDS with SLR is 0.86, 0.80, and 0.75 for CSR, GFZ, and JPL, respectively. The correlation coefficients also demonstrate that our new calibration method ensures accurate $C_{20}$ estimations for HUST-Grace2023.

Apart from the $C_{20}$ results, the other low-degree coefficients derived from HUST also have a good agreement with those derived from the other SDSs. Especially, the $C_{30}$ estimations for gravity field official solution centers, including HUST, is slightly overestimated, compared with those derived from SLR. The overestimated $C_{30}$ is related to the satellite internal temperature stability and the sun/eclipse exposure changes, which are advisable to replace by the SLR estimation (Kvas et al., 2019). As for $C_{21}$ and $S_{21}$, the results of HUST are consistent with those of CSR and JPL. In addition, for $C_{21}$, $S_{21}$, $C_{22}$ and $S_{22}$, the general consistency is also observed between GRACE and GRACE-FO estimations.

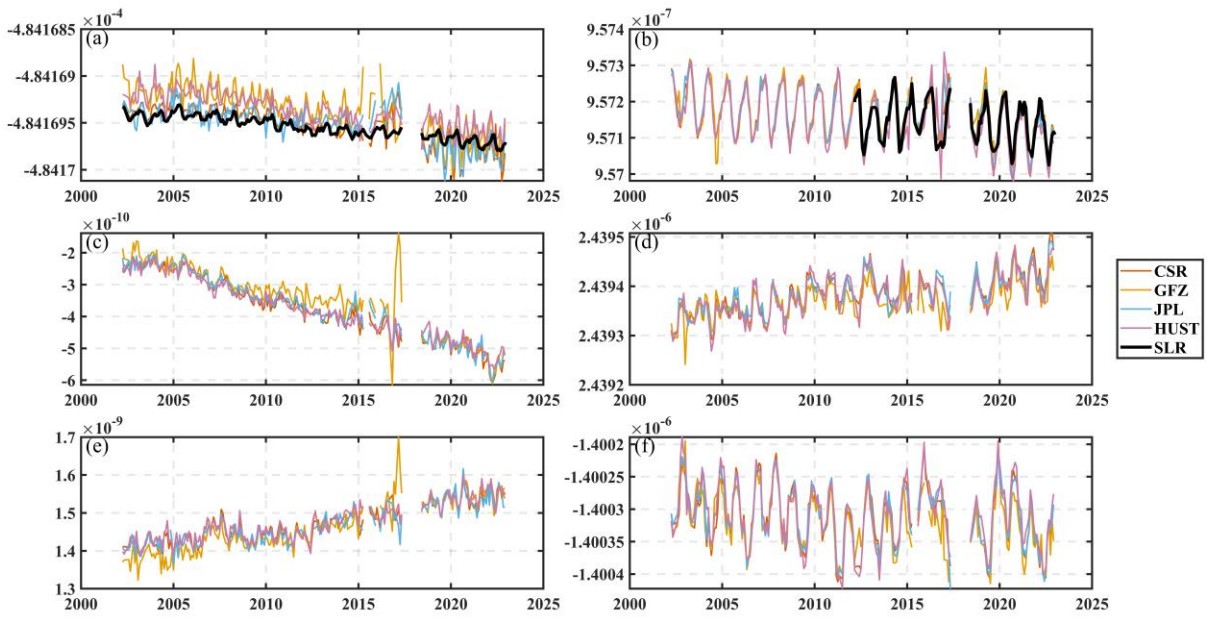

**Figure 7 Geopotential coefficients derived from SLR, CSR, GFZ, JPL and HUST. The geopotential coefficients include $C_{20}$ (a), $C_{30}$ (b), $C_{21}$ (c), $C_{22}$ (d), $S_{21}$ (e) and $S_{22}$ (f) during 2002-2022. Note that the SLR curve is only shown in (a) or (b).**

**Table 3 The correlation coefficient along with SLR computed by $C_{20}$ derived from different science data system (SDSs).**

**Note that the result of SLR is derived from TN-14 document (Loomis et al.,2020).**

| SDS | SLR | CSR | GFZ | JPL | HUST |
|-----|-----|-----|-----|-----|------|
| SLR | 1.00 | 0.86 | 0.80 | 0.75 | 0.86 |
| CSR | 0.86 | 1.00 | 0.75 | 0.87 | 0.81 |
| GFZ | 0.80 | 0.75 | 1.00 | 0.75 | 0.74 |
| JPL | 0.75 | 0.58 | 0.75 | 1.00 | 0.70 |
| HUST | 0.86 | 0.81 | 0.74 | 0.70 | 1.00 |

**Table 4 Cumulative geoid height differences (cm) of CSR, GFZ, JPL and HUST-Grace2024 model. All models are truncated up to degree and order 96.**

| Time | ①CSR | ②GFZ | ③JPL | ④HUST | $\frac{④-①}{①}$ | $\frac{④-②}{②}$ | $\frac{④-③}{③}$ |
|------|------|------|------|-------|------|------|------|
| 2009.01 | 0.64 | 0.77 | 0.87 | 0.59 | -7.7% | -22.5% | -31.9% |
| 2006.08 | 0.70 | 1.01 | 0.97 | 0.56 | -21.0% | -45.0% | -42.6% |
| 2010.10 | 0.67 | 0.74 | 0.85 | 0.53 | -19.7% | -27.9% | -37.0% |

| 2005 to 2010 | 0.75 | 0.98 | 1.00 | 0.66 | -12.8% | -33.2% | -34.7% |
| 2019.10 | 1.31 | 2.35 | 1.91 | 0.98 | -25.4% | -58.3% | -48.9% |
| 2022.05 | 1.25 | 2.00 | 1.65 | 0.91 | -27.4% | -54.3% | -44.7% |
| 2018 to 2022 | 1.27 | 2.10 | 1.78 | 0.99 | -21.7% | -52.9% | -44.1% |

### 3.1.2 Analysis in Spatial Domain

In order to evaluate the performance of our HUST-Grace2024 model in the spatial domain, the global mass changes are calculated in terms of equivalent water heights (EWHs) with a resolution of 1°, smoothed by a 300 km Gaussian filter and a P3M6 decorrelation filter. Based on the global EWHs, the commonly used spatial criterions are then computed, including annual amplitudes, yearly trends and root mean squares (RMSs) of temporal signal residuals. Here, the annual amplitudes and yearly trends are used to assess the performance of retrieving temporal signals, while the RMSs are used for the temporal noise evaluation. As shown in Equation (13), the RMSs are calculated on the basis of the terrestrial water storage anomalies (TWSA) residuals, which have removed the yearly trends, semi-annual amplitudes and annual amplitudes from the original time series.

$$RMS = rms(TWSA_{total} - TWSA_{trend} - TWSA_{annual-amplitude} - TWSA_{semiannual-amplitude}) \qquad (13)$$

The global annual amplitudes, yearly trends for different GRACE solutions during 2005 to 2016 are displayed in Figure 8, while those for the GRACE-FO solution are shown in Figure 9. The annual amplitude for HUST-Grace2024 presents a good agreement with the official solutions in the areas of South America, medium Africa, and the south of Asia, ensuring the ability to capture large-scale mass change over the world. In addition, the RMS value for the result without any filter is also plotted as shown in Figure 10 and Figure 11. In order to mitigate the signal leakage from land into the ocean, we make a 400 km buffer zone from the coastline for the results. The temporal noise in HUST-Grace2024 is significant smaller than the solutions derived from JPL and GFZ, while it's comparable with the CSR result generally. Indeed, we don't account the impact of co-seismic changes and post- seismic changes during the computation of TWSA residual. The effects of seismic changes are local and not significant in the TWSA residual RMS distribution, which may be a little beyond the research scope of the paper.

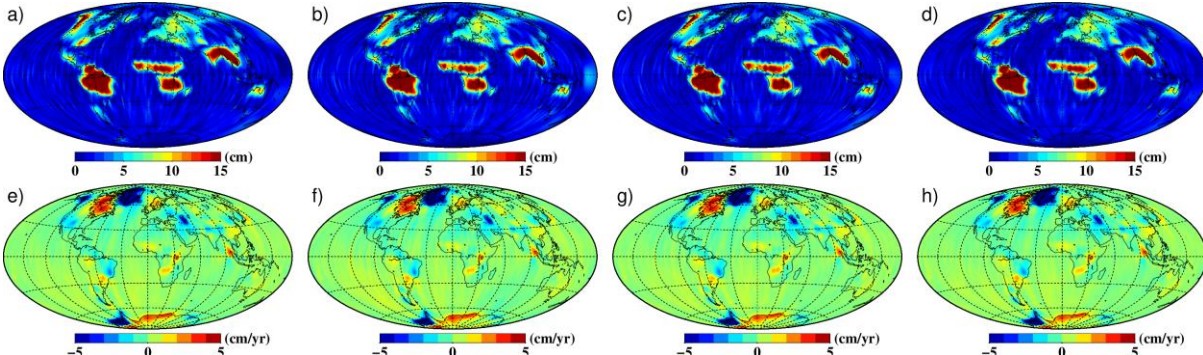

**Figure 8 Global annual amplitudes (row 1), yearly trends (row 2) during 2005-2016 derived from different products including (a, e) CSR RL06, (b, f) GFZ RL06, (c, g) JPL RL06 and (d, h) HUST-Grace2024. Note that all results are smoothed by 300km Gaussian filter and P3M6 decorrelation filter.**

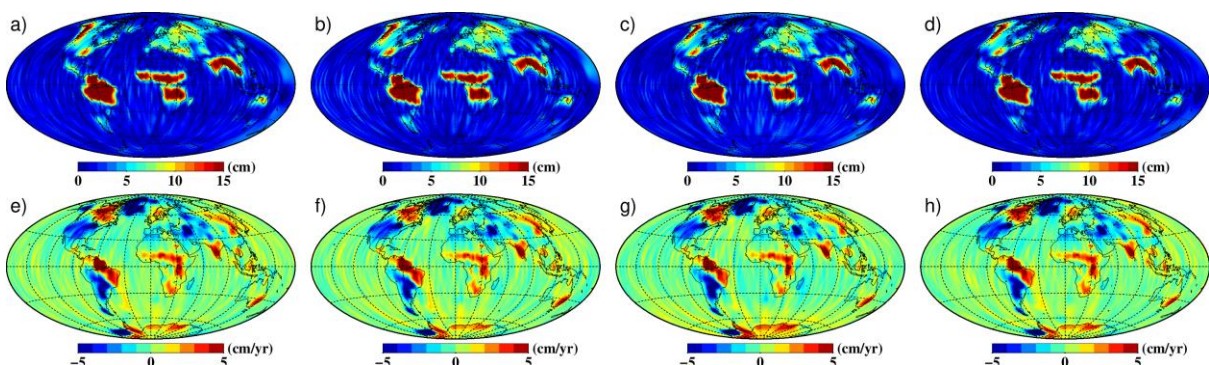

**Figure 9 Same as Figure 8 but for GRACE-FO solutions during 2018 to 2022.**

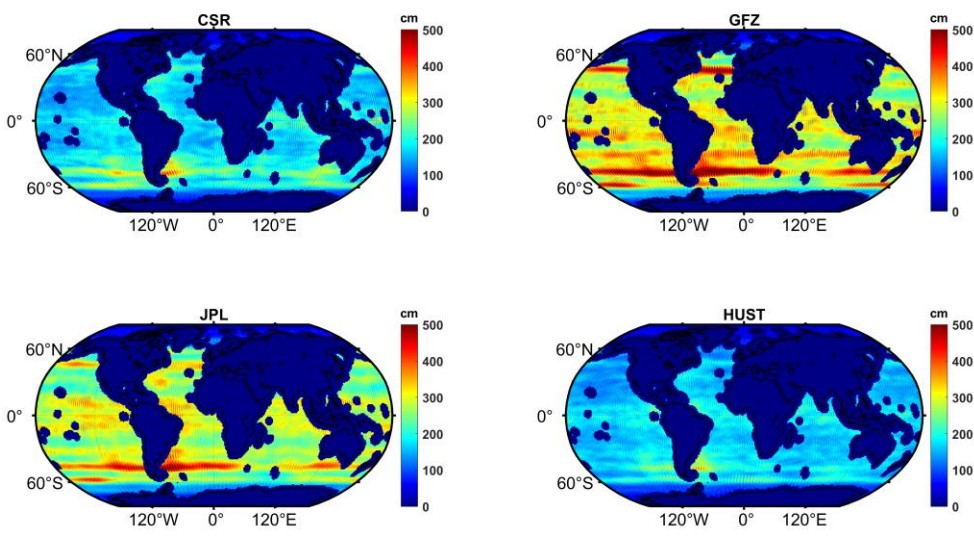

**Figure 10 the RMS value of EWH residual which has been removed the climatology part (bias, amplitude, trend) for different GRACE solutions during 2005-2016 over the open ocean with a 400 km buffer zone from the coastline. Note**

10 **that none filter is applied here.**

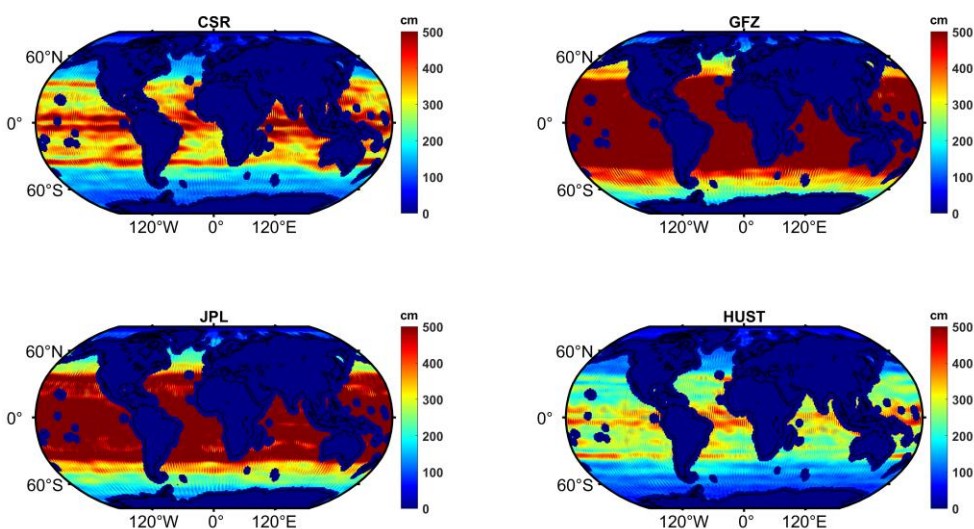

**Figure 11 Same as Figure 10 but for GRACE-FO solutions during 2018-2022.**

In order to assess the noise level of our HUST-Grace2024 model, we select an open ocean area as shown in Figure 10 or Figure 11, which is considered to have the smallest temporal signal remaining and is very suitable for the temporal gravity field assessment in spatial domain.

The EWH residual is computed by the EWHs (without any filter) minus their climatology part (bias, amplitude, and trend) based on the temporal gravity field derived from CSR RL06, JPL RL06, GFZ RL06, and HUST-Grace2024 during 2005 to 2016. As shown in Figure 12, in most months during 2005 to 2016, the residuals over a selected open ocean in our model are significantly smaller than those from JPL and GFZ, while being comparable with those from CSR. The RMS values of the residuals are 132.3 cm, 228.1 cm, 209.2 cm, and 118.0 cm for CSR RL06, GFZ RL06, JPL RL06, and HUST-Grace2024, respectively. In addition, in our GRACE-FO solution, the residual EWH over a selected open ocean is further reduced when compared with the latest official solution RL06.1. The RMS value of the residual is 235.2 cm, 488.6 cm, 390.3 cm, and 177.7 cm for CSR RL06.2, JPL RL06.1, GFZ RL06.1, and HUST-Grace2024, respectively. The GRACE-FO solution indicates that our hybrid processing can significantly reduce the temporal noise over the open ocean when compared to the GRACE solution.

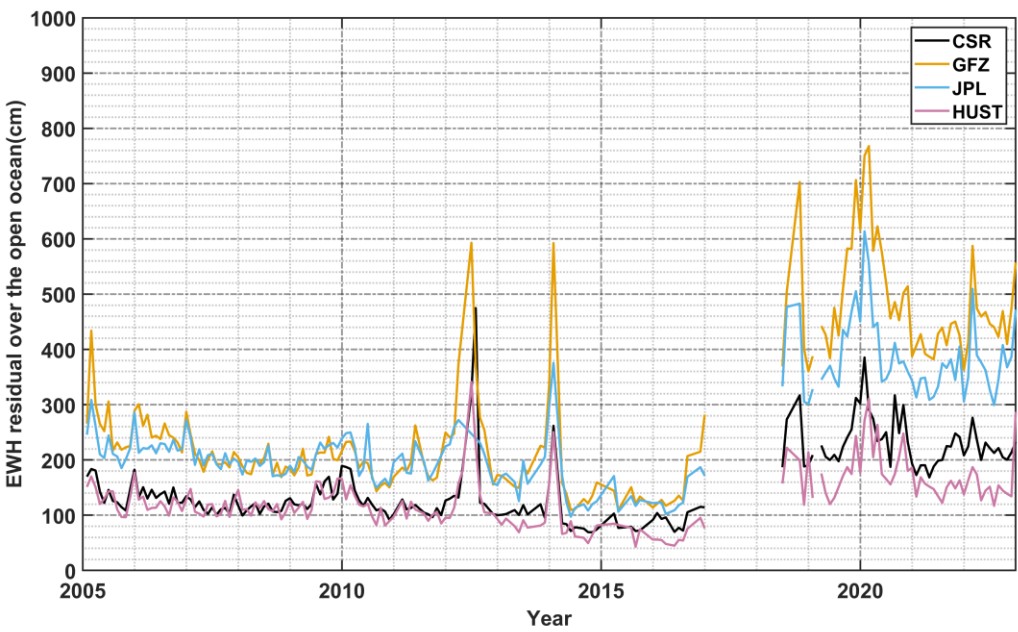

**Figure 12 Equivalent Water Height (EWH) residual over a selected open ocean series during 2005 to 2016 and 2018 to 2022. The residual is computed by the EWH (without any filter) minus its climatology part (bias, amplitude, and trend).**

For a more quantitative comparison of the noise level between HUST product and the latest official GRACE-FO solutions, we calculate the weighted mean of the RMS EWH residual over the representative deserts and the east of Antarctic, which is assumed to have an extremely small temporal signal remaining. Due to the fact that the climatology part (bias, trend, and amplitude) has been removed from the residual result, the residual can be regarded as a mis-modelling signal or temporal noise that remained in the model solution. As listed in Table 5, in the Sahara desert, the residual for our solution is 0.90 cm but 0.98 cm and 1.45 cm for CSR and GFZ, indicating the noise reduction of our GRACE-FO solution is 8.3% and 37.8% when compared to CSR and GFZ solution. In addition, the noise reductions of our HUST-Grace2024 model become more significant when none filter is applied. For instance, the reductions over Sahara Desert reach to 22.3% and 68.1% when compared to CSR and GFZ solution. The similar phenomenon is also observed over the east of Antarctic and the Thar desert.

**Table 5 Weighted mean EWH residual (cm) over the representative deserts and the east of the Antarctic. Note that the result without any filter and 300 km Gaussian filter are computed. The location of assessment area of the representative deserts and the east of the Antarctic are summarized as East Antarctic (74°S~80°S, 60°E~120°E), Sahara Desert (28°N~34°N, 3°W~9°E), Thar Desert (23°N~30°N, 70°E~75°E)**

| Radius | Regions | ①CSR | ②GFZ | ③HUST | $\frac{③-①}{①}$ | $\frac{③-②}{②}$ |
|--------|---------|------|------|-------|------------------|------------------|
| 0km | East Antarctic | 4.79 | 9.76 | 4.00 | -16.5% | -59.0% |

| | | | | | | |
|---|---|---|---|---|---|---|
| | Sahara Desert | 21.41 | 52.27 | 16.62 | -22.3% | -68.1% |
| | Thar Desert | 15.65 | 31.98 | 11.48 | -26.6% | -64.1% |
| | East Antarctic | 0.85 | 0.80 | 0.75 | -12.3% | -7.0% |
| 300km | Sahara Desert | 0.98 | 1.45 | 0.90 | -8.3% | -37.8% |
| | Thar Desert | 3.41 | 3.67 | 3.33 | -2.3% | -9.3% |

In order to assess the performance of our model in retaining temporal signal, we compute the annual amplitudes over 48 representative river basins. The locations of these 48 representative river basins are same as the Fig. S3 in Zhou et al. (2018). Following Scanlon et al. (2016) and Zhou et al. (2018), the scatters of the annual amplitudes for 48 river basins are plotted for the cross-comparison between different GRACE-FO solutions (Figure 13). The GRACE-FO solutions including CSR RL06.2, GFZ RL06.1, JPL RL06.1 and HUST-Grace2024. Theoretically, when the slope of the fitted line (the black dash line in Figure 13) is closer to 1, the temporal signals derived from different models are more similar. Compared with HUST-Grace2024, the scatters are close to the fitted line, and the slope is 1.00, 1.03, and 1.00 for CSR RL06.1, GFZ RL06.1, and JPL RL06.1, respectively. It demonstrates the general good agreements between our model and three official models in retrieving basin-scale temporal signals.

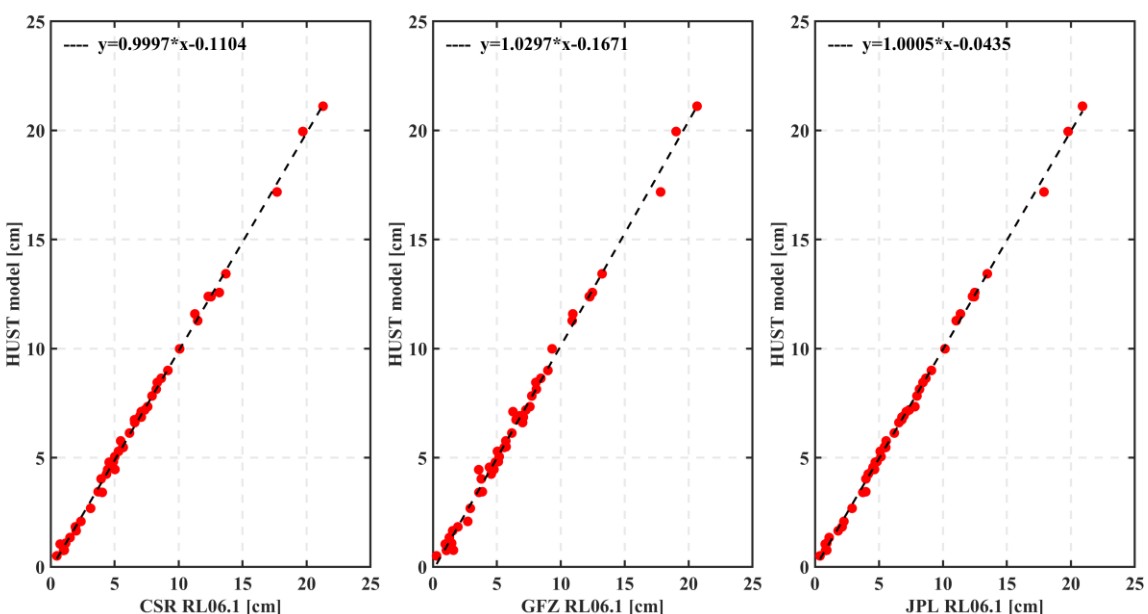

**Figure 13 The annual amplitudes derived from CSR RL06.2, GFZ RL06.1, JPL RL06.1 and HUST-Grace2024 in GRACE-FO solution during 2018-2022. The x-axis value stands for the amplitude from official solution, while the y-axis stands for the value from HUST solution. The black dash line is linear fit line.**

For a more quantitative analysis, we computed the terrestrial water storage anomaly (TWSA) variation time series following the equation (13). As shown in Figure 14, generally, the TWSA time series computed by our HUST-Grace2024 model is consistent with the latest GRACE-FO solution in 8 representative basins such as the Amazon, Mekong, Ob, and so on. These 8 basins are selected based on different latitudes for a general comparison. The

comparison results demonstrate our HUST-Grace2024 model has almost the same ability to maintain temporal signal when compared with the latest official GRACE-FO solutions. To make a more general comparisons, as listed in Table 6, we also computed the TWSA residuals over the 48 representative basins. In addition, we also apply a high pass filter to the TWSA residuals in order to mitigate the inter-annual in the TWSA residuals. In the large-scale basins like the Amazon, the RMSs are 2.64 cm, 3.14 cm, 2.91 cm, and 3.12 cm for HUST-Grace2024,

CSR RL06.2, GFZ RL06.1, JPL RL06.1 respectively. For the small-scale basin such as Orange, the RMSs are 1.49 cm, 1.63 cm, 1.62 cm, and 1.60 cm for HUST-Grace2024, CSR RL06.2, GFZ RL06.1, and JPL RL06.1, respectively. In addition, when compared with official GRACE-FO solutions, HUST-Grace2024 model presents the smallest RMSs over 68% of these basins. The comparison results also demonstrate the general better performance of our HUST-Grace2024 model in reducing temporal noise when compared to the official GRACE-

FO solutions.

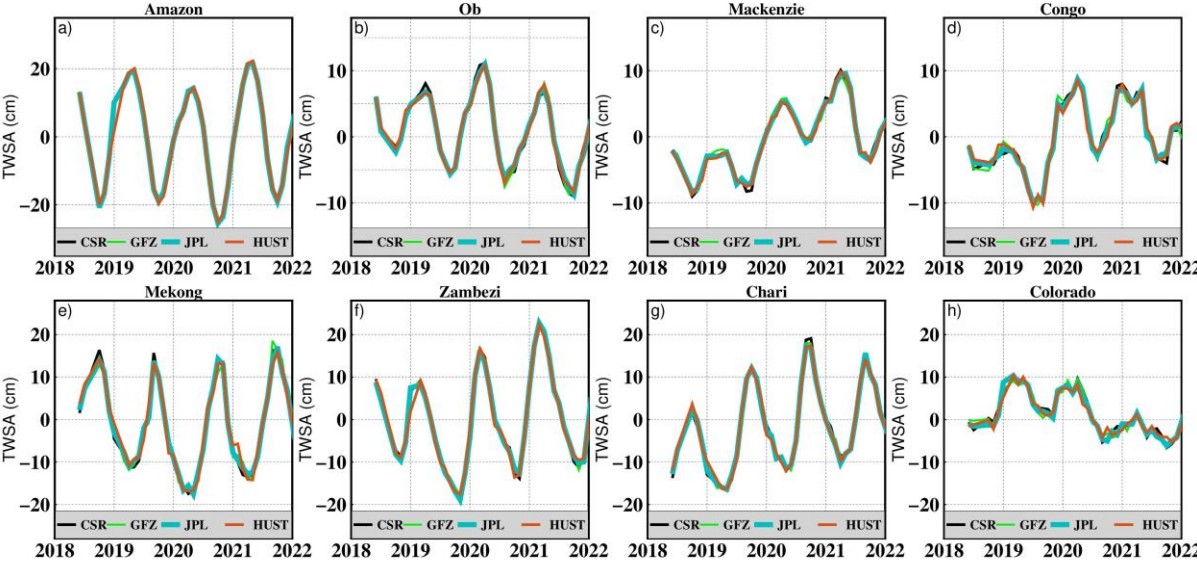

**Figure 14 Terrestrial water storage anomaly (TWSA) variation time series of different basins during 2018 to 2022 which are derived from latest official GRACE-FO solution and HUST-Grace2024.**

**Table 6 RMSs (cm) of 48 representative basins derived from latest official GRACE-FO solution and HUST-Grace2024. The bold value represents the minimum RMS. Note that all RMS value has been processed by a high pass filter to mitigate the inter-annual signal in the EWH residual.**

| ID | Basin Name | CSR | GFZ | JPL | HUST |
|----|-----------|------|------|------|------|
| 1 | Amazon | 3.14 | 2.91 | 3.12 | **2.64** |
| 2 | Congo | 3.03 | 3.09 | 2.94 | **2.87** |
| 3 | Mississippi | 2.39 | **2.32** | **2.32** | **2.32** |
| 4 | Ob | **1.09** | 1.17 | 1.20 | 1.16 |
| 5 | Parana | 3.23 | 3.29 | 3.21 | **2.99** |
| 6 | Nile | 2.41 | 2.61 | 2.54 | **2.31** |
| 7 | Yenisei | **1.18** | 1.35 | 1.22 | 1.21 |
| 8 | Lena | 1.54 | 1.56 | 1.56 | **1.49** |
| 9 | Niger | 1.68 | 1.71 | **1.63** | 1.68 |
| 10 | Amur | **1.57** | 1.74 | 1.70 | 1.64 |
| 11 | Yangtze | 3.22 | 3.41 | 3.21 | **3.20** |
| 12 | Mackenzie | 2.99 | 2.98 | 3.05 | **2.90** |
| 13 | Volga | 1.64 | **1.52** | 1.55 | 1.70 |
| 14 | Zambezi | 3.86 | **3.77** | 3.84 | 3.81 |
| 15 | Lake Eyre | **1.39** | 1.45 | 1.66 | 1.50 |
| 16 | Nelson | 2.40 | 2.33 | 2.37 | **2.25** |
| 17 | St-Lawrence | 2.81 | 2.83 | 2.73 | **2.71** |
| 18 | Murray | 2.18 | 2.30 | 2.33 | **2.05** |
| 19 | Ganges | 3.27 | **2.90** | 3.10 | 3.14 |
| 20 | Orange | 1.63 | 1.62 | 1.60 | **1.49** |
| 21 | Indus | 2.73 | 2.63 | 2.73 | **2.59** |
| 22 | Chari | 2.83 | 2.90 | 2.59 | **2.52** |
| 23 | Orinoco | 4.80 | 5.00 | 5.07 | **4.72** |
| 24 | Tocantins | 5.26 | 5.16 | **4.94** | 5.31 |

| | | | | | |
|---|---|---|---|---|---|
| 25 | Yukon | 1.31 | **1.25** | 1.34 | 1.34 |
| 26 | Danube | **2.01** | 2.34 | 2.15 | 2.09 |
| 27 | Mekong | **3.86** | 4.24 | 4.05 | 4.24 |
| 28 | Okavango | 4.04 | 3.81 | 4.04 | **4.02** |
| 29 | Huang He | 1.52 | 1.78 | **1.51** | 1.52 |
| 30 | Euphrates | 3.79 | **3.53** | 3.57 | 3.78 |
| 31 | Jubba | 2.85 | 2.64 | **2.51** | 3.05 |
| 32 | Columbia | 1.51 | 1.53 | 1.43 | **1.39** |
| 33 | Brahmaputra | 2.43 | **2.30** | 2.34 | 2.35 |
| 34 | Kolyma | **1.30** | 1.35 | 1.35 | 1.55 |
| 35 | Colorado | 2.48 | 2.23 | 2.27 | **2.14** |
| 36 | Rio Grande | 2.29 | 2.27 | 2.20 | **1.97** |
| 37 | Sao Francisco | 4.48 | **3.81** | 4.38 | 4.12 |
| 38 | Dniepr | **2.02** | 2.18 | 2.19 | 2.35 |
| 39 | Amu Darya | **1.38** | 1.54 | 1.50 | 1.54 |
| 40 | Limpopo | 3.10 | 3.08 | 3.10 | **3.02** |
| 41 | Senegal | **1.92** | 2.32 | 2.27 | 2.07 |
| 42 | Tarim | **0.94** | 1.10 | 1.02 | 1.06 |
| 43 | Don | 2.79 | 2.81 | 2.55 | **2.52** |
| 44 | Syrdarya | 1.25 | 1.47 | 1.27 | **1.20** |
| 45 | Xi | 3.67 | 3.93 | 3.75 | **3.63** |
| 46 | Volta | 2.56 | 3.04 | 2.73 | **2.29** |
| 47 | Northern Dvina | 1.59 | **1.53** | 1.58 | 1.73 |
| 48 | Khatanga | **1.29** | 1.54 | 1.36 | 1.48 |

Overall, our HUST-Grace2024 model has good performance in both spatial and spectral domain when compared to the latest official solutions. It presents the balance between maintaining temporal signal and reducing temporal noise. In addition, the result related to GRACE-FO indicates the good performance of our hybrid processing chain

in determining the temporal gravity field models for this mission, and it even presents more notable improvements when compared with the official GRACE solutions.

### 3.1.3 Comparing with HUST-Grace2020

In order to make a quantitative assessment for the contribution of different strategies in our hybrid processing chain, we make a comparison between HUST-Grace2024 and previous version HUST-Grace products such as HUST-Grace2020. The comparison mainly focuses on the contribution of newest AOD1B product AOD1B RL07 and the hybrid algorithm developed in HUST-Grace2024. Due to HUST-Grace2020 only spans from January 2003 to July 2016, the comparison will only focus on GRACE solutions except for GRACE-FO solutions.

The details of processing strategies for different HUST products can summarized as Table 7. It's noted that only main diagonal scale factor elements are estimated during HUST-Grace2020 determination. The difference between HUST-Grace2020 and HUST-Grace2024prelimary can mainly be accounted for the new stochastic model. And the difference between HUST-Grace2024prelimary and HUST-Grace2024 can mainly be accounted for the new input AOD1B products. It should keep in mind that the GRACE solution derived from CSR has the same fully scale factor matrix with HUST-Grace2024prelimary, which can also make a assessment for the contribution of new stochastic model.

**Table 7 Different processing details between HUST-Grace2020 and HUST-Grace2024**

|  | HUST-Grace2020 | HUST-Grace2024prelimary | HUST-Grace2024 |
|---|---|---|---|
| Fully scale factor matrix | ✗ | ✓ | ✓ |
| Newest AOD1B product | ✗ | ✗ | ✓ |
| New Stochastic Model | ✗ | ✓ | ✓ |

For an assessment for our strategies adopted in hybrid processing chain, we select two monthly products derived from different GRACE observation time shown in Figure 15 and Figure 16, which can help us demonstrate the contribution of newest AOD1B product or our refined processing chain mentioned in the paper. The result derived from September 2009 can be regarded as a "good" observation time period, when the GRACE satellites are both in well temperature control. However, the result derived from April 2012 is usually regarded as a "bad" or "not so good" observation time period, when the GRACE-B lost its temperature control from now on.

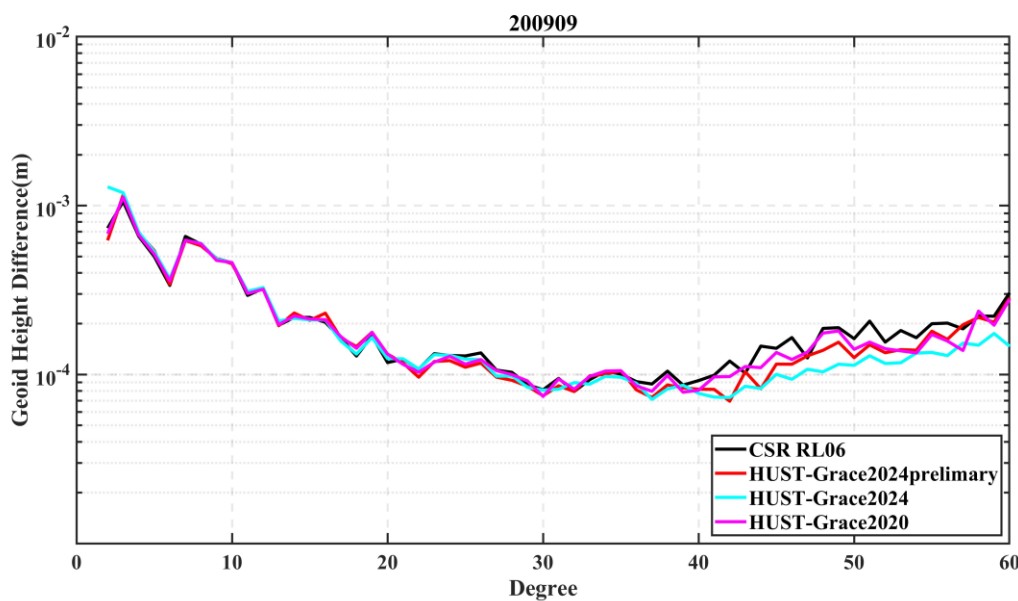

**Figure 15 the Geoid height difference per degree for different GRACE solutions in September 2009**

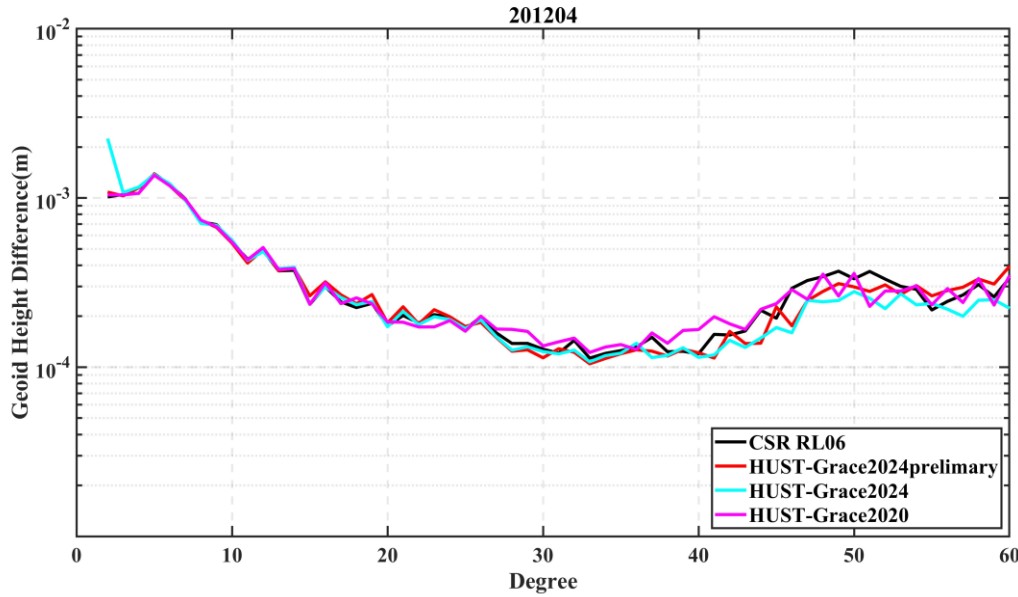

**Figure 16 the Geoid height difference per degree for different GRACE solutions in April 2012**

5    As shown in Figure 15, the geoid height difference derived from different GRACE solutions are very similar below degree 40. The geoid height difference derived from HUST-Grace2020 is generally smaller than that derived from CSR, which mainly demonstrate that there is less temporal noise in HUST-Grace2020. The HUST-Grace2024prelimary has smaller geoid height difference than that from HUST-Grace2020 between degree 40 and degree 50, which indicates that our new stochastic model has a positive effect on reducing high frequency noise.

10   It's should be noted that the adoption of AOD1B RL07 has positive effect on reducing aliasing error during the

temporal gravity field determination, especially over degree 40. However, as shown in Figure 16, the geoid height difference is generally larger than that shown in Figure 15, which demonstrates that bad temperature control on GRACE-B has negative impact on temporal gravity field determination. The geoid height difference derived from HUST-Grace2020 is slightly larger than that derived from CSR between degree 25 and degree 40, and is comparable with that from CSR between degree 40 and degree 60. While it's noticeable that the geoid height difference derived from both HUST-Grace2024prelimary and HUST-Grace2024 is smaller than that from CSR between degree 40 and degree 60. The result indicates that the new stochastic model is still beneficial for the temporal gravity field determination during "not so good" GRACE observation time period.

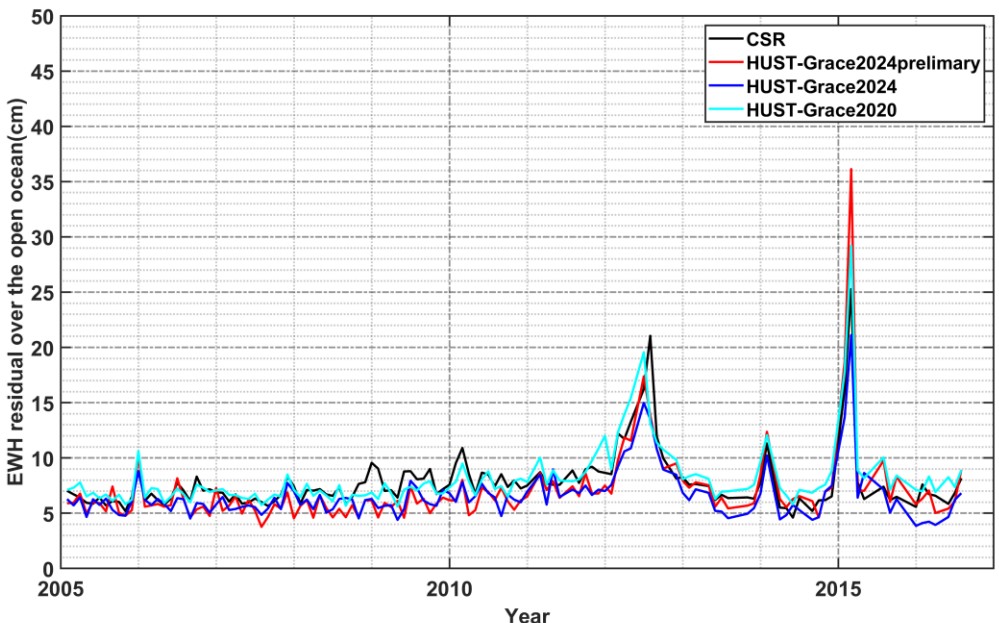

**Figure 17 Equivalent Water Height (EWH) residual over a selected open ocean series during 2005 to 2016. The residual is computed by the EWH (without any filter) minus its climatology part (bias, amplitude, and trend).**

Similar to the approach mentioned in the paper above, we still take the open ocean residual as the criteria for the assessment of temporal noise in gravity field products. As shown in Figure 17, the temporal noise in HUST-Grace2024prelimary is comparable with that in HUST-Grace2024 as well as HUST-Grace2020 during 2005 and 2010, however, the temporal noise in HUST-Grace2024 is significant smaller than those in other temporal gravity field products. The RMS value is 8.23 cm, 7.76 cm, 6.94 cm, 8.59 cm for CSR, HUST-Grace2024prelimary, HUST-Grace2024, HUST-Grace2020, respectively. Comparing with HUST-Grace2020, the reduction of temporal noise is 9.6%, 19.2% for HUST-Grace2024prelimary and HUST-Grace2024.

**4 Conclusion**

On the basis of a new hybrid processing chain, we developed a new series of temporal gravity field models HUST-Grace2024. The highlights of this hybrid processing chain include the improved GRACE and GRACE-FO L1B datasets, the updated background models, the new pre-processing chain, the hybrid data weighting method, and the updated accelerometer scale factor matrix.

During determining our HUST-Grace2024 temporal gravity field model, the latest GRACE and GRACE-FO L1B datasets are used, including RL03 data for GRACE and RL04 data for GRACE-FO. The background force models are also updated, including the static gravity field GOCO06s, and the newest atmosphere and ocean de-aliasing product AOD RL07. Meanwhile, to comprehensively consider the color noise at different frequency, the empirical kinematic parameters as well as the stochastic model are applied as a hybrid weighting method. The hybrid weighting method present notable improvements when compared to the traditional weighting method. For the accelerometer scale factor matrix, we fully considered the misalignment between SRF and AF as well as the non-orthogonality of three AF axes. The scale factor estimations at the main diagonal line are more stable for GRACE mission than GRACE-FO mission, especially for the along-track direction. In contrast, the scale factor estimations off the main diagonal line are similar between GRACE and GRACE-FO, which reflects the similar performance of alignment or orthogonality of the axes related to accelerometers.

In the hybrid processing chain, we specially developed a hybrid outliers detection method for the pre-processing procedure. The method is based on the SOE file and the inter-satellite pointing angles. Based on several practical computations for the real GRACE observations, it is advisable to set the threshold of pointing angles at 50 mrad and a 100 second margin to detect outliers in range rate observations, while setting the threshold at 20 mrad and a 600 second margin for accelerometer data. The updating pre-processing strategy ensures us to accurately exclude outliers in observations on one hand, and improve the quality of temporal gravity field estimations on the other hand.

Further, the newly developed temporal gravity field model HUST-Grace2024 is compared with the latest official solutions, including CSR RL06, GFZ RL06, and JPL RL06 for GRACE (RL06.1 for GRACE-FO derived from GFZ and JPL, while RL06.2 for GRACE-FO derived from CSR). In the spectral domain, the cumulative geoid height difference during 2005 to 2016 reflects the lower noise level of HUST-Grace2024, with a noise reduction

of 3.0% to 27.2% when compared to the official GRACE solutions. The more notable noise reductions are observed for GRACE-FO solutions, which vary from 17.6% to 46.9%. The results present the better efficiency of our hybrid processing chain for GRACE-FO mission than GRACE mission. In the spatial domain, a noticeable temporal noise reduction is observed over open ocean in HUST-Grace2024. The mean RMSs over the open ocean

are 132.3 cm, 228.1 cm, 209.2 cm, and 118.0 cm for CSR RL06, JPL RL06, GFZ RL06 and HUST-Grace2024, respectively. As for GRACE-FO solutions, they are respectively 235.2 cm, 488.6 cm, 390.3 cm, and 177.7 cm . The annual amplitudes of 48 major basins derived from HUST-Grace2024 show good agreement with those derived from the official solutions, while over 68% basins present the smallest RMSs when compared to the official solutions. The comparisons in the spectral and spatial domains indicate that our updated hybrid process

chain can result in better GRACE and GRACE-FO monthly gravity field estimations.

Overall, the updating pre-processing method gives us an insight into observation data, especially excluding some outliers during the satellite onboard events. Meanwhile, the hybrid weighting method leads to a better understanding of frequency dependent noise in the real GRACE and GRACE-FO observations. As described in Flechtner et al. (2016), Chen et al. (2020) and Zhou et al. (2023), the force model errors and accelerometer errors

have been the dominant limiting factors for GRACE-FO. Our hybrid processing chain can help us understand the stochastic characteristics of instrument noise and develop a series of temporal gravity fields based on the observation data from GRACE-FO.

**Author contribution**

Zhicai Luo reviewed the article. Zebing Zhou reviewed the article. Xiang Guo reviewed the article. Yaozong Li performed the formal analysis and visualization. Hao Zhou and Lijun Zheng performed the HUST-Grace2024 gravity field processing and prepared the manuscript with contributions from all co-authors.

**Data Availability Statement**

The GRACE Level 1B data in this study are freely available at freely available at http://isdcftp.gfz-potsdam.de (Wen et al., 2010), and the kinematic orbits are provided by ITSG at http://ftp.tugraz.at (Suesser-Rechberger et al., 2019). HUST-Grace2024 model is available at https://doi.org/10.5880/ICGEM.2024.001

**Competing interests**

The authors declare that they have no conflict of interest.

**Acknowledgments**

This research was funded by the National Natural Science Foundation of China (No. 42074018, 41931074, 42061134007) and the National Key Research and Development Program of China (No. 2023YFC2907003). The computation is completed in the HPC Platform of Huazhong University of Science and Technology.

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
