# Peer review of "HUST-Grace2024: a new GRACE-only gravity field time series based on more than 20 years satellite geodesy data and a hybrid processing chain"

_Earth System Science Data, 2024_

## Author Comment (AC1)

**AC: We thank the reviewers for taking time to review this manuscript. Their insightful remarks have helped us to identify parts in the manuscript which needed clarification and certainly allowed us to improve the quality of this paper. A version of the manuscript with markup which shows the differences between the initial submission and the revised version where the reviewer comments are addressed, is attached to this document.**

**Anonymous Referee #2**

Review about the paper

**HUST-Grace2024: a new GRACE-only gravity field time series based on more than 20 years satellite geodesy data and a hybrid processing chain**

submitted to Earth System Science Data (https://doi.org/10.5194/essd-2024-39)

Authors: Hao Zhou, Lijun Zheng, Yaozong Li, Xiang Guo, Zebing Zhou, and Zhicai Luo

**General Remarks:**

This study develops a new series of temporal gravity field models HUSTGrace2024, on the basis of a hybrid data processing chain and updated GRACE and GRACE-FO L1B datasets. The hybrid processing chain takes account of the newest atmosphere and ocean de-aliasing product (AOD1B RL07), an improved data weighting method and the updated accelerometer scale factor matrix, which leads to a better temporal gravity field series. Comparison with the current official GRACE and GRACE-FO solutions (released by CSR, GFZ and JPL) demonstrates that the HUSTGrace2024 models have a lower noise level (with a reduction of 10% to 30%) and comparable signal amplitudes in both the long-term trend and seasonal variations.

The datasets developed by the authors are useful to those who investigate the global climate change and geodynamics with satellite gravimetry. Meanwhile, the hybrid data processing chain provides insights into more accurate determination of the global time-variable gravity field models. In particular, the detailed parameter setting suggestions in GRACE and GRACE-FO L1B data processing (such as the thresholds to detect outliers in range-rate observations, and to set the margin for accelerometer datasets, etc.) are beneficial to more reliable gravity field modeling. I would like to recommend minor revisions of the manuscript before publication in Earth System Science Data, according to the comments as follows.

**AC: We thank the reviewer for their insightful comments, which helped us to identify parts in need of clarification and undoubtedly allowed us to improve the quality of the manuscript. Below is the point-by-point response to the specific remarks.**

1. On the selection of a quiet ocean area to assess the noise level

In Section 3.1.2, the authors select an open ocean area (in the middle Pacific Ocean) to evaluate the noise level of the datasets (Page 21, Lines 4-5; Page 22, Figure 9). However, the location of the selected ocean area (as well as its latitude and longitude ranges) is not described in the text. Please clarify it.

In addition, using only parts of the ocean is probably not the best way to assess the noise level of the GRACE and GRACE-FO temporal gravity field series. I suggest the authors to add the results for the global ocean. To reduce the effect of signal leakage from the land areas, a buffer zone (with 400 or 500 km from the coastlines) can be considered when determining the global ocean coverage. The RMS comparison for the global ocean among the different products would be more convincing to evaluate the noise level.

**AC: Thank you for insightful comments, indeed the using only parts of the ocean isn't the best way to assess the noise level of the GRACE and GRACE-FO temporal gravity field series based on our HUST-Grace2024 products comparing with the official products. So, we follow your suggestion, recompute the RMS residual over the open ocean (with a 400 km buffer zone from the coastline). In addition, we also add the location of the representative desert and east of the Antarctic in the text. The location of the assessment area can be founded in the note of Table 4 (Page 22)**

**Here is the modified note according to Table 4:**

**"Weighted mean EWH residual (cm) over the representative deserts and the east of the Antarctic. Note that the result without any filter and 300 km Gaussian filter are computed. The location of assessment area of the representative deserts and the east of the Antarctic are summarized as East Antarctic (74°S~80°S, 60°E~120°E), Sahara Desert (28°N~34°N, 3°W~9°E), Thar Desert (23°N~30°N, 70°E~75°E)"**
* * *
2. On the RMSs of TWSA residuals to assess the noise level

In Section 3.1.2, the RMSs of TWSA residuals (by removing the yearly trends and annual amplitudes from the original time series) are used to assess the noise level (Page 19, Equation 13). Nevertheless, removing only the yearly trends and annual amplitudes is likely not enough to separate the signal and noise. For example, in the regions of South America (Amazon basin), East Africa (Nile basin) and South Asia (the surroundings of Bengal Bay), there are notable effects in the RMSs (see the row 3 in Figures 6 and 7), which are probably due to the inter-annual changes in the TWSA. Thus it might be better to take into account the inter-annual variations in Equation (13), with a high-pass filter (for instance, with a threshold of 1.5 or 2 years) to mitigate the inter-annual impact.

In addition, the co- and post-seismic changes of several gigantic earthquakes during the GRACE observation period may also affect the RMSs results to some extent. Since the seismic effects are local and not significant in the RMS distribution (Figure 6), adding some discussions to acknowledge this issue should be enough.

Another concern is about the semi-annual changes in the GRACE and GRACE-FO time series. Normally both the annual and semi-annual terms are included to represent the seasonal variations. So, I would suggest the authors to consider the semi-annual term in Equation (13).

**AC: Thank you for insightful comments, it really helps us to improve our research work furtherly. During the computation of TWSA residual, we actually have removed the trend, annual amplitude, semi-annual term part from the original TWSA, and we have modified the Equation (13) according to your valuable comments. About the co- and post-seismic changes in the TWSA residual, we don't make a quantitative assessment for these changes in the TWSA actually, and we also add some necessary discussion about this in the text according to your comment. About the inter-annual TWSA changes, we also apply a high pass filter to the residual, replot the figure in the paper.**

**Modified sentence**

**".... Indeed, we don't account the impact of co-seismic changes and post-seismic changes during the computation of TWSA residual. The effects of seismic changes are local and not significant in the TWSA residual RMS distribution, which may be a little beyond the research scope of the paper."**

**"... the RMSs are calculated on the basis of the terrestrial water storage anomalies (TWSA) residuals, which have removed the yearly trends, semi-annual amplitudes and annual amplitudes from the original time series."**

3. On the assessment of low degree coefficients C20 and C30

In Section 3.1.1, the authors compare the datasets in the spectral domain. From Figure 5 one can find that the low degree coefficients are notably different between the ones from HUST and those from the three official solutions. Therefore, it should be better to add a specific comparison for the low-degree coefficients (e.g., C20 and C30). By comparing the time C20 and C30 time series among the HUST, CSR (GFZ and JPL) and those from the SLR observations (i.e., TN-14; see Loomis et al., 2020), the advantage of the data product in this study may be better demonstrated.

**AC:Thank you for your insightful comments, the difference of low degree coefficients from different temporal gravity field products may be due to different accelerometer calibration strategy adopted by the solution centers. And we have added the comparison results in Section 3.1.1.**

4. On the acknowledgement of the difference in data inputs between HUST and the official solutions

The authors compare the HUST solutions and those from CSR, GFZ and JPL, and find that their solution performs better than the others. However, one should keep in mind that the data inputs are different for the solutions. The HUST solutions use the newest atmosphere and ocean de-aliasing product (AOD1B RL07), whereas the CSR (GFZ and JPL) solutions use the old version of the AOD1B product. In the near future, CSR (GFZ and JPL) will release the GRACE and GRACE-FO RL07 solutions, which are comparable (in the data inputs) to the HUST solutions. Whether the HUST datasets perform better than the RL07 official products remains a question. Please add some discussions to acknowledge this issue.

In addition, CSR has released the RL06.2 solutions for GRACE-FO since September 2023. But the authors do not mention this official product. Please add some necessary statements or discussions.

**AC: Thank you for this valuable suggestion. Indeed, as your statement in the comment, the different input products will have a positive effect on our solutions. In order to demonstrate our solutions performance, we compare our HUST-Grace2024 with the previous solution, such as HUST-Grace2020, which use the same AOD1B products as inputs with the official products and some other control parameter experiments. All the comparison results have been written in Section 3.1.3. About the newest version products from CSR, we have replaced the results from RL06.1 to RL06.2 for CSR GRACE-FO solution.**
* * *
(1) Page 1, Line 25: singal -> signal

**AC: Thank you for careful observation, we have corrected this typo.**

**Modified sentence:**

**"… noise level and remains consistent amplitudes over 48 basins in signal content compared with the official…"**
* * *
(2) Page 4, Line 1: the quiet ocean -> a selected open ocean region

For the "quiet ocean" in the rest parts of the manuscript (e.g., Page 21, Lines 4, 10 and 13; Page 22, Line 2; Page 28, Line 3; …), it should be better to change it into "open ocean".

**AC: Thank you for your suggestion, we have rephased the sentence according to your advice. Since there are many sentences to be rephased, please refer to the revised manuscript.**

(3) Page 4, Line 7: background model -> background models

**AC: Thank you for your careful observation, we have rephased this title in our revised paper.**

**Modified title:**

**"Updating of GRACE data and background models"**
* * *
(4) Page 6, Line 5: Importance of renewing -> Renewing of

To be consistent with the title of Section 2.1 (Page 4, Line 7).

**AC: Thank you for your really careful observation, we have rephased this title in our revised paper.**

**Modified title:**

**"Renewing of HUST-GRACE2024 data pre-processing strategy"**
* * *
(5) Page 16, Lines 20-21: The result indicates that our proposed hybrid weighting approach can obtain the better temporal gravity field solutions.

This statement is not appropriate for the Methods section. Please remove the sentence and consider to make this statement in the Results or Discussion section.

**AC: Thank you for your valuable suggestion, we have removed this sentence in our revised paper.**

(6) Page 17, Line 2: from 2005 to 2010 -> from 2005 to 2010 for GRACE solutions

**AC: Thank you for your careful observation, we have modified this sentence in our revised paper.**

**Modified sentence:**

**"… from 2005 to 2010 for GRACE solutions and 2018 to 2022 for GRACE-FO solutions …"**
* * *
(7) Page 17, Line 21: The numerical values "7.528, 9.827, 10.049, and 6.561", and many other values in the rest parts of the text (e.g., Page 19, Lines 17 and 19; Page 20. Lines 1 and 2; …), as well as the values in the tables (e.g., Tables 3 to 5)

It does not make much practical sense to use 3 decimal places for the numerical values in the results. Using 1 decimal place (or at most 2) should be enough.

**AC: Thank you for your suggestion, we use the proper decimal place for our numerical values according to your advice in our revised paper.**
* * *
(8) Page 19, Line 4: a decorrelation filter

Which decorrelation filter? Please specify it.

**AC: Thank you for your comment, we have modified the sentence in our revised paper.**

**Modified sentence:**

**"… Gaussian filter and a P3M6 decorrelation filter"**
* * *
(9) Page 21, Line 15: "and HUST-Grace2024" -> "and HUST-Grace2024, respectively"

**AC: Thank you for your comment, we have modified the sentence in our revised paper.**

**Modified sentence:**

**"… GFZ RL6.1, and HUST-Grace2024, respectively"**
* * *
(10) Page 22, Lines 5-6: the representative deserts and the east of Antarctic

Please describe the latitude and longitude ranges for these selected areas, or show their coverages in a map.

**AC: Thank you for your comment, please see our previous comment about this question.**
* * *
(11) Page 23, Line 13: annual amplitude -> annual amplitudes

**AC: Thank you for your comment, we have modified the sentence in our revised paper.**

**Modified sentence:**

**"The annual amplitudes derived …"**
* * *
(12) Page 26, Line 2: official solution -> official solutions

**AC: Thank you for your comment, we have modified the sentence in our revised paper.**

**Modified sentence:**

**"… to the latest official solutions …"**
* * *
(13) Page 30, Line 27, and Page 31, Line 1:

The references Cheng & Ries (2017) and Cheng & Ries (2023) are not mentioned in the text.

**AC: Thank you for your comment, we have added some comparison results about the low degree coefficients in section 3.1.1, which may contain the references mentioned in your comments in our revised paper.**
* * *
(14) Page 35, Line 8: "Geo. Jou. Int."

Please use "Geophys. J. Int.", or use the complete journal name.

**AC: Thank you for your comments, we have modified this error in our revised paper.**

---

## Author Comment (AC2)

**AC: We thank the reviewers for taking time to review this manuscript. Their insightful remarks have helped us to identify parts in the manuscript which needed clarification and certainly allowed us to improve the quality of this paper.**

**Anonymous Referee #3**

Review about the paper

**HUST-Grace2024: a new GRACE-only gravity field time series based on more than 20 years satellite geodesy data and a hybrid processing chain**

submitted to Earth System Science Data (https://doi.org/10.5194/essd-2024-39)

Authors: Hao Zhou, Lijun Zheng, Yaozong Li, Xiang Guo, Zebing Zhou, and Zhicai Luo

**General Remarks:**

The manuscript provides detailed information about their new version of the time-variable gravity field series using a new data processing strategy and new input data. However, the following are my primary comments and suggestions for major comments to the study.

**AC: We thank the reviewer for their insightful comments, which helped us to identify parts in need of clarification and undoubtedly allowed us to improve the quality of the manuscript. Below is the point-by-point response to the specific remarks.**

1.The GRACE result section should incorporate also comparisons with previous versions of HUST-Grace to assess how they correspond with the official GRACE solutions.

**AC: Thank you for insightful comments, Reviewer #2 has the same comments with you, we have added some comparison results in the new section. The new section is about comparing with our solution with previous versions of HUST-Grace such as HUST-Grace2020 to assess our newest solution performance.**
* * *
2 During the GRACE mission, particularly after 2010, more factors emerged, including maneuvers and the GRACE-B battery issue. Hence, it is imperative to incorporate the GRACE time period post-2010 within the results section, as it will serve as a comprehensive testing phase for the new step procedures provided in the study.

.

**AC: Thank you for insightful comments, it really helps us to improve our research work furtherly. We have added some GRACE results for the time period post-2010 according to your valuable comments. Please refer to the modified result section in the revised paper.**
* * *
3.The study's proposed improvement steps should clearly and prominently show the impact of either the new accelerometer calibration or the new AOD1B product. This can be done in the result section, comparing it with the prior version of HUST solutions.

**AC:Thank you for your valuable comments, we have added some comparison results in the section 3.1.3. Please refer to the revised paper.**
* * *
4.The metrics used to examine the models in the comparison section were used in a very subset of the significance of the magnitudes. Although the comparison is intended to emphasize that the HUST-

Grace2024 model is better in a meaningful way, it may not be meaningful if the cm or mm orders are 3 digits finer after the comma. For example, Page 19, Line 17, RMSs over ocean is in cm, but the notations are well below mm.

**AC: Thank you for this careful observation, Reviewer #2 has some similar comments with you. We have modified this error in our revised paper.**

---

## Author Comment (AC3)

**AC: We thank the reviewers for taking time to review this manuscript. Their insightful remarks have helped us to identify parts in the manuscript which needed clarification and certainly allowed us to improve the quality of this paper.**

**Anonymous Referee #1**

Review about the paper

**HUST-Grace2024: a new GRACE-only gravity field time series based on more than 20 years satellite geodesy data and a hybrid processing chain**

submitted to Earth System Science Data (https://doi.org/10.5194/essd-2024-39)

Authors: Hao Zhou, Lijun Zheng, Yaozong Li, Xiang Guo, Zebing Zhou, and Zhicai Luo

**General Remarks:**

The manuscript outlines a data processing strategy, yielding impressive improvements in its recovered temporal gravity solutions

**AC: We thank the reviewer for their insightful comments, which helped us to identify parts in need of clarification and undoubtedly allowed us to improve the quality of the manuscript. Below is the point-by-point response to the specific remarks.**

1.Please specify the time interval used for constructing the observation equation, considering the differing sampling rates between the kinematic orbit (10 seconds) and other L1B data (5 seconds). Additionally, please elaborate on the error assessment strategy for kinematic orbits, including the criteria

for error identification, and whether interpolated epochs are included in constructing the observation equation.

**AC: Thank you for insightful comments. During the HUST-Grace2024 temporal gravity field determination, the integration time interval is 5 seconds and the original observation equation is build based on 5 seconds for orbit and range-rate observation. As your comments stressed, the kinematic sample rate is different from the other L1B data and we simply truncate the original observation equation for integration orbit according to the GPS time tag in the kinematic observation. Actually, during the kinematic preprocessing, we use the reduced dynamic orbit as the criteria for error identification, and when the difference between the reduced dynamic orbit and the kinematic orbit exceeds 20 cm, we will give a quality flag to the kinematic orbit at a specific GPS time and will not use the kinematic observation for the temporal gravity field determination later on. As for the gap in the kinematic observation, we fill the gap by zero value and don't use the observation to construct the observation equation.**
* * *
2. Please quantitatively analyze the accuracy improvement of the temporal gravity recovery by kinematic orbit and GNV 1B.

**AC: Thank you for insightful comments, it really helps us to improve our research work furtherly. We have added some comparison results for kinematic orbit and GNV 1B. Please refer to the modified result section in the revised paper.**
* * *
3.Is the influence of the thruster accounted for in preprocessing? If so, please clarify whether THR 1B data is utilized and provide details regarding the number of epochs affected by thruster start-up time.

**AC:Thank you for your valuable comments, actually we don't account the influence of the thruster in our HUST-Grace2024 processing. However, we also do some experiment about this influence magnitude on temporal gravity field determination (not shown in the paper) and the following figure is our experiment result. The experiment is designed as: (1) Finding a thruster start-up time tag in the THR 1B data (2) Building a margin time interval at 0 second, 1 second, 5 second and 10 second based on the thruster start-up time tag, regarding the thruster start-up time tag as a center time tag. (3) Removing the thruster active accelerometer observation value from the original observation, and the gap due to thruster active is filled by the interpolated value. Generally, the thruster last less than 1 second, and we think the thruster has little effect in our HUST-Grace2024 data processing.**

[Figure]

**4.** I would like to know the performance of your products in the later stage of GRACE. Please extend the time period for comparing GRACE results from 2005-2010 to 2005-2015.

**AC: Thank you for this useful suggestion, Reviewer #3 has some similar comments with you. We have added some comparison result in our revised paper.**

———————————————————————————————

5. Correct a typographical error on Page 12, Line 18, where 'equation (9)' should be amended to 'equation (6)'.

**AC: Thank you for this careful observation, we have corrected this error in our revised paper.**

———————————————————————————————

6. On Page 20, Line 7, consider rephrasing "indicating a reduction of -12.8%, -33.2%, and -34.7%" for clarity.

**AC: Thank you for this careful observation, we have corrected this error in our revised paper.**

**Modified sentence:**

**"... indicating an average cumulative geoid height difference reduction of -12.8%, -33.2%, and -34.7%…"**

---

## Author Comment (AC4)

**AC: We thank the reviewers for taking time to review this manuscript. Their insightful remarks have helped us to identify parts in the manuscript which needed clarification and certainly allowed us to improve the quality of this paper.**

**Community Comment#2**

Review about the paper

**HUST-Grace2024: a new GRACE-only gravity field time series based on more than 20 years satellite geodesy data and a hybrid processing chain**

submitted to Earth System Science Data (https://doi.org/10.5194/essd-2024-39)

Authors: Hao Zhou, Lijun Zheng, Yaozong Li, Xiang Guo, Zebing Zhou, and Zhicai Luo

1. Page 20, L7, 'including (a, e, f) CSR RL06, (b, f, j) GFZ RL06, (e, g, k) JPL RL06 and (d, h, f) HUST-Grace2024' should be corrected as 'including (a, e, i) CSR RL06, (b, f, j) GFZ RL06, (c, g, k) JPL RL06 and (d, h, l) HUST-Grace2024'

**AC: Thank you for insightful comments, we have replotted the figure in our revised paper.**

---

## Author Comment (AC5)

**AC: We thank the reviewers for taking time to review this manuscript. Their insightful remarks have helped us to identify parts in the manuscript which needed clarification and certainly allowed us to improve the quality of this paper.**

**Community Comment#1**

Review about the paper

**HUST-Grace2024: a new GRACE-only gravity field time series based on more than 20 years satellite geodesy data and a hybrid processing chain**

submitted to Earth System Science Data (https://doi.org/10.5194/essd-2024-39)

Authors: Hao Zhou, Lijun Zheng, Yaozong Li, Xiang Guo, Zebing Zhou, and Zhicai Luo

**General Remarks:**

The improvement relative to official gravity solutions is impressive. The improvement may come from 1) new accelerometer product, 2) new AOD1B product, 3) algorithm in the manuscript. I would like to see how much of the improvement comes from the new products, and how much comes from the algorithm, thus consolidating the contribution of this work. I suggest the authors add such a controlled variable experiment.

**AC: We thank the reviewer for their insightful comments, which helped us to identify parts in need of clarification and undoubtedly allowed us to improve the quality of the manuscript. Below is the point-by-point response to the specific remarks. Indeed, as you stress in the comments, HUST-Grace2024 may be benefit from the hybrid processing chain including many refinements mentioned in the paper. In order to make a quantitative assessment for the refinements, we compare**

**our products with prior version products such as HUST-Grace2020 in both spectral and spatial domain. Please refer to section 3.1.3 in the revised paper.**

1. Fig. 8, the HUST result is better over the CSR result globally, with the exception in western Pacific. Is there a reason for this?

**AC: Thank you for insightful comments, the reason maybe come from the different choice of ocean tide models. Actually, according to the suggestion from the reviewers, it's more reasonable to compare the temporal noise between different temporal gravity field based on the EWH residual over the open ocean. We have replotted the figure in our revised paper.**

2. Page 13, L18, a typo in 'shown'.

**AC: Thank you for insightful comments, we have modified the typo in our revised paper.**